# Dual roles of microbes in mediating soil carbon dynamics in response to warming

Shuqi Qin[1,2], Dianye Zhang[1,2], Bin Wei[1,2,3] & Yuanhe Yang ®[1,2,3] ✉

Understanding the alterations in soil microbial communities in response to climate warming and their controls over soil carbon (C) processes is crucial for projecting permafrost C-climate feedback. However, previous studies have mainly focused on microorganism-mediated soil C release, and little is known about whether and how climate warming affects microbial anabolism and the subsequent C input in permafrost regions. Here, based on a more than half-decade of in situ warming experiment, we show that compared with ambient control, warming significantly reduces microbial C use efficiency and enhances microbial network complexity, which promotes soil heterotrophic respiration. Meanwhile, microbial necromass markedly accumulates under warming likely due to preferential microbial decomposition of plant-derived C, further leading to the increase in mineral-associated organic C. Altogether, these results demonstrate dual roles of microbes in affecting soil C release and stabilization, implying that permafrost C-climate feedback would weaken over time with dampened response of microbial respiration and increased proportion of stable C pool.

Permafrost, defined as subsurface earth materials that remain at or below 0 °C for two or more years, covers approximately 15% of the global land area[1,2]. Low temperatures and frozen conditions in permafrost regions have constrained microbial utilization of soil organic carbon (SOC), and preserved a large carbon (C) stock over long-term scale. It has been estimated that $1014^{+194}_{-175}$ Pg C (1 Pg = $10^{15}$ g) is stored in the top 3 m soils across permafrost region in the Northern Hemisphere[3], accounting for more than 30% of the global soil C stock[4]. This vast C pool has recently been subjected to net loss[5-7], since permafrost regions are warming twice to four times faster than the global average in recent decades[2,8]. The alleviation of temperature constraint would promote microbial metabolic processes, which may result in increased soil C release in the form of C dioxide ($CO_2$) and methane ($CH_4$) and further trigger a positive C-climate feedback[4,9]. Given that soil C cycling is largely determined by microbial activities, and that explicit representation of microbial processes in Earth system models could improve model performance[10,11], a thorough understanding of microbial responses to climate warming is crucial for more accurate predictions of permafrost C fate as well as the associated C-climate feedback.

It is increasingly recognized that soil microorganisms could play two contrasting roles in mediating terrestrial C cycling: microbial catabolic activities lead to soil C release to the atmosphere, whereas microbial anabolism generates a set of products that contribute to the stable soil C pool[12,13]. Both of these two microbial processes would be affected by climate warming[14,15], with the direction and magnitude of their changes determining soil C dynamics in a warmer world. Warming-accelerated microbial decomposition rate could lead to soil C loss[16], while warming-induced accumulation/reduction of microbial-derived compounds such as microbial necromass C would promote/weaken soil C stabilization over long term[12]. However, our understanding of microbial roles in permafrost region is mainly confined to microorganism-mediated soil C release[17-20], with less attention being paid to microbial anabolism and the resulted C input under warming scenario. In fact, in addition to the energy-yielding metabolisms, warming could also affect other physiological attributes (e.g., growth, turnover[21] and C use efficiency (CUE)[22]) related to microbial anabolism and deposition of anabolic products. These attributes could reflect microbial controls over soil C storage[12] and are vital parameters for soil

[1]State Key Laboratory of Vegetation and Environmental Change, Institute of Botany, Chinese Academy of Sciences, 100093 Beijing, China. [2]China National Botanical Garden, 100093 Beijing, China. [3]University of Chinese Academy of Sciences, 100049 Beijing, China. ✉e-mail: yhyang@ibcas.ac.cn

C models[23]. Nevertheless, it remains unclear whether and how climate warming alters soil microbial physiology thereby affecting microbial-derived C in permafrost regions, which greatly restricts model projections of soil C dynamics. Comprehensive evaluation of microbial mediation over soil C processes considering both microbial catabolism and anabolism under climate warming is thus urgently needed to constrain model parameters so as to accurately predict the direction and strength of permafrost C-climate feedback.

The Tibetan Plateau is known as the "Third Pole" of the world. An area of $1.06 \times 10^6$ km² of the plateau is underlain by permafrost, representing ~75% of mountain permafrost across the Northern Hemisphere[24,25]. Widespread climate warming has occurred across the plateau over recent decades, with a rate of 0.5–0.7 °C decade⁻¹ since the 1980s[26]. In such a warming context, an accumulation of SOC was observed in the subsurface soils (10–30 cm) across the plateau, while no significant change was detected for the topsoils[27]. Yet to date, microbial mechanisms underlying soil C dynamics are largely unexplored on the plateau. To fill this knowledge gap, here we provided thorough experimental evidence about microbial responses to experimental warming, and illustrated microbial roles in affecting soil C cycling based on an in situ open-top chamber (OTC) warming experiment on the north-eastern plateau[28]. We compared a suite of topsoil (0–10 cm) microbial attributes between warming and control plots after 6 years of continual warming treatment, including microbial community composition, metabolic capacities, physiology (growth, turnover and CUE), and necromass C. We then discussed microbial mediation over two soil C cycling processes in response to warming, i.e., soil C release characterized by heterotrophic respiration ($R_h$) and soil C stabilization reflected by mineral-associated organic C (MAOC). We hypothesized that in this temperature-constrained permafrost ecosystem, experimental warming would significantly alter microbial community composition and elevate metabolic capacities, which led to the increased $R_h$. We also hypothesized that the warming treatment would inhibit microbial CUE but stimulate turnover, promoting the accumulation of microbial-derived C and thus MAOC. Our results

revealed that experimental warming did not alter the overall microbial community composition, but decreased microbial CUE and structured more complex network, which was responsible for the increased $R_h$. In addition, microbial necromass C accumulated over time probably due to preferential microbial decomposition of plant-derived compounds, further contributing to the increased proportion of MAOC. These results collectively demonstrate dual microbial roles in affecting soil C processes in response to experimental warming in this permafrost ecosystem.

## Results

### Warming effects on microbial community composition and network

The continual OTC warming treatment significantly increased topsoil temperature by 0.8–2.2 °C since 2014 ($P < 0.05$; Supplementary Table 1). It is expected that experimental warming-induced alterations in soil temperature, plant biomass and edaphic properties (Supplementary Tables 1 and 2) would change soil microbial community. To test this prediction, we determined prokaryotic (i.e., bacterial and archaeal) and fungal community composition using amplicon sequencing of 16S rRNA gene and ITS2 region, respectively. In contrast to our expectation, the results showed that the warming treatment did not alter the alpha diversity (including richness and Shannon–Wiener index) of prokaryotes and fungi (Supplementary Fig. 1). Meanwhile, the relative abundance of dominant microbial taxa at the phylum level was similar between warming and control conditions, although some taxa at lower taxonomic level were differentially abundant (Supplementary Fig. 2; see Supplementary Note 1 for details). The overall microbial community composition was not significantly altered by experimental warming according to non-parametric multivariate statistical tests (ANOSIM, Adonis and MRPP; Supplementary Table 3).

Despite the weak response of microbial community composition, molecular ecological networks based on Pearson correlations of log-transformed sequence data differed between the warming and control treatments (Fig. 1). Compared with the control, the warming treatment

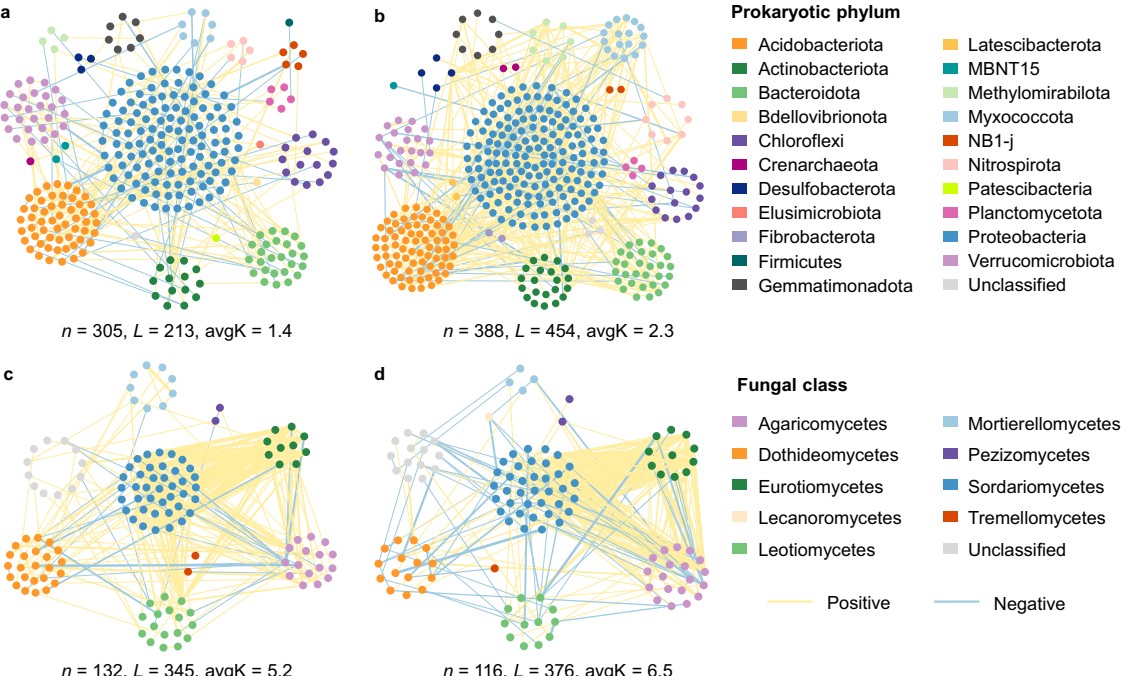

**Fig. 1 | Co-occurrence patterns of topsoil microbial communities as affected by experimental warming.** Visualization of prokaryotic (**a**, **b**) and fungal (**c**, **d**) networks under control (**a**, **c**) and warming (**b**, **d**) conditions. Nodes in the network denote individual ASVs whose color indicates taxonomic groups. Lines between the nodes represent significant correlations, with yellow and blue indicating positive and negative correlation, respectively. Line width is proportional to the strength of the relationship. *n* number of total nodes, *L* number of total links, avgK average degree. Source data are provided as a Source Data file.

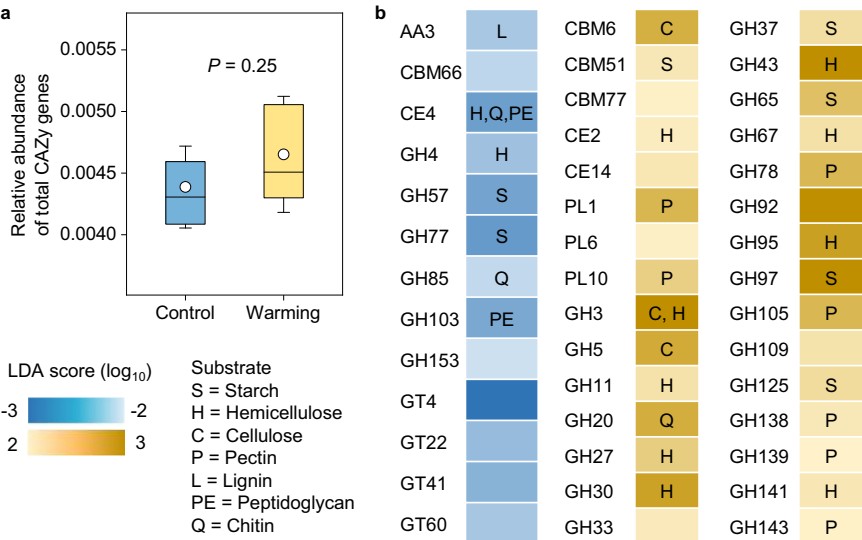

**Fig. 2 | Effects of experimental warming on relative abundance of CAZy genes in topsoil metagenomes. a** Comparison of relative abundance of total carbohydrate-active enzymes (CAZy) genes between warming and control conditions according to paired samples *t*-test (two-sided). Box represents the interquartile range, with blue and yellow indicating control and warming, respectively. Horizontal line and circle within the box show the median and mean value, respectively. The whisker denotes SD (*n* = 10, independent samples). **b** Heatmap showing the linear discriminant analysis (LDA) scores computed for CAZy gene families differentially abundant (LDA score >2, unadjusted *P* < 0.05) between warming and control conditions. Families enriched under control condition are indicated by a negative LDA score (blue), and the warming treatment enriched families are indicated by a positive score (yellow). AA auxiliary activity, CBM carbohydrate-binding module, CE carbohydrate esterase, GH glycoside hydrolase, GT glycosyltransferase, PL polysaccharide lyase. Letters above the heatmap represent the substrate category utilized by enzymes from the corresponding CAZy gene family. Source data are provided as a Source Data file.

had less amplicon sequence variant (ASV) numbers used for network construction, but showed larger final network size for prokaryotes as reflected by total nodes (Fig. 1a, b and Supplementary Table 4). Prokaryotic network under warming also exhibited higher connectivity (total links), average connectivity (avgK; average links per node), and average clustering coefficient (the extent of node clustering) (Supplementary Table 4). In addition, experimental warming increased total links and avgK of fungal network (Fig. 1c, d), and also elevated relative modularity and the proportion of keystone nodes for both prokaryotic and fungal networks (Supplementary Fig. 3). These results collectively indicated altered network structure and enhanced network complexity by the warming treatment. Further analyses revealed that the composition of prokaryotic and fungal communities detected in the network markedly differed between the warming and control treatments (*P* < 0.001; Supplementary Table 3), which was partly driven by the altered edaphic (soil temperature, pH, and $NO_3^-$-N content) and plant (Normalized Difference Vegetation Index (NDVI) and root biomass) variables under warming (Supplementary Fig. 4a, b). Moreover, the connections between network module-based eigengenes and environmental variables varied between the warming and control treatments (Supplementary Fig. 4c–f). For more details about network descriptions, keystone node identification and module-eigengene analysis, see Supplementary Note 2.

## Impacts of experimental warming on microbial metabolic capacities

We examined microbial C-utilization potential based on metagenomic sequencing and annotation against the carbohydrate-active enzymes (CAZy) database. Among the genes annotated, glycoside hydrolases (GHs) and glycosyltransferases (GTs) were the most abundant CAZy classes under both control and warming conditions, while the least abundant classes were polysaccharide lyases (PLs) and auxiliary activities (AAs). In total, we detected 237 CAZy families, including 114 GHs, 46 GTs, 40 carbohydrate-binding modules (CBMs), 14 carbohydrate esterases (CEs), 18 PLs, and 5 AAs. The relative abundance of the total CAZy gene families remained unaltered under warming (Fig. 2a).

Nevertheless, 30 gene families primarily associated with the degradation of plant-derived C (such as starch, hemicellulose, cellulose and pectin) were enriched under warming, while 13 gene families were more abundant under control as revealed by linear discriminant analysis (LDA) effect size (LEfSe) method (Fig. 2b). Consistent with the results based on CAZy database, gene annotation against the Kyoto Encyclopedia of Genes and Genomes (KEGG) database also revealed that experimental warming elevated the relative abundance of genes involved in the degradation of starch, hemicellulose and cellulose (Supplementary Fig. 5). These results collectively indicated enhanced potential for utilization of plant-derived C under warming conditions.

## Effects of experimental warming on microbial physiology

Considering the importance of microbial physiology in affecting soil C response to warming, we adopted a substrate-independent method based on [18]O incorporation into microbial DNA to determine gross microbial growth rate[29], concurrent with which microbial respiration rate was measured. Considering that physiological processes operate per unit of microbial biomass[29], mass-specific rates were used for comparison. The results showed that in situ warming altered some of microbial physiology attributes: mass-specific growth rate remained unchanged between warming and control conditions (Fig. 3a), while respiration rate was significantly accelerated by the warming treatment (*P* < 0.05; Fig. 3b). Consequently, a lower microbial CUE (i.e., the proportion of assimilated C used for growth) was detected under warming (warming vs. control: 34 ± 2% vs. 41 ± 2%; hereafter, data are reported as means ± standard errors; *P* < 0.01; Fig. 3c). Moreover, microbial turnover rate did not differ between warming and control conditions (Supplementary Fig. 6a).

## Microbial necromass C and soil C dynamics in response to warming

To evaluate warming effects on microbial-derived C, we used a widely accepted biomarker, i.e., amino sugars, to trace microbial necromass[30]. Three individual amino sugar (including glucosamine (GluN), galactosamine (GalN), and muramic acid (MurA)) were acquired. The

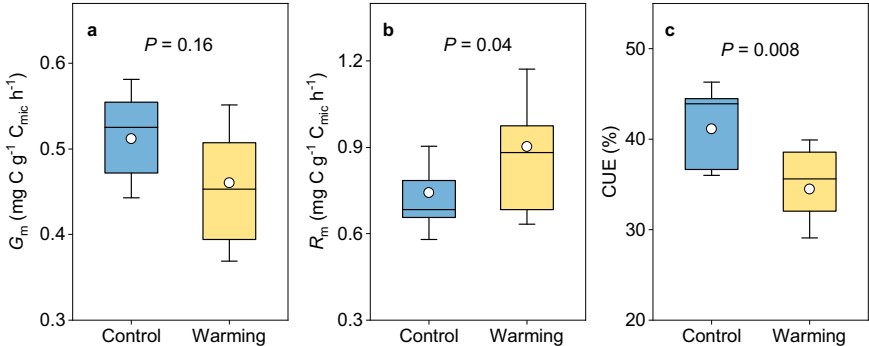

**Fig. 3 | Responses of microbial physiology to experimental warming.** Comparison of microbial mass-specific growth ($G_m$; **a**), mass-specific respiration ($R_m$; **b**), and carbon use efficiency (CUE; **c**) for topsoil under warming and control conditions. Box represents the interquartile range, with blue and yellow indicating control and warming, respectively. Horizontal line and circle within the box show the median and mean value, respectively. The whisker denotes SD ($n = 10$, independent samples). Paired samples $t$-tests (two-sided) were conducted to compare means of $G_m$, $R_m$, and CUE between the warming and control treatments. Source data are provided as a Source Data file.

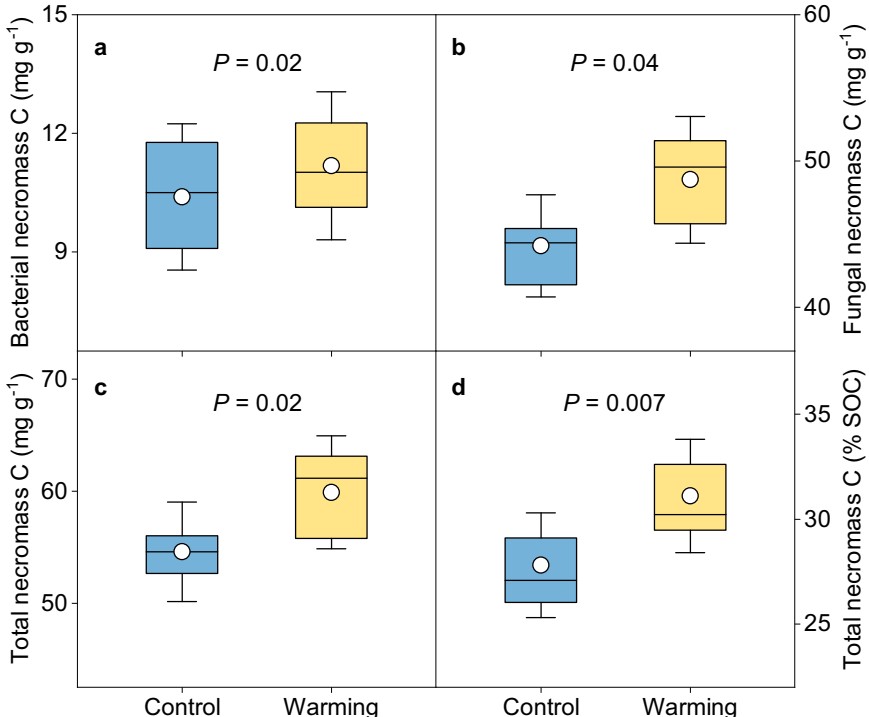

**Fig. 4 | Effects of experimental warming on topsoil microbial necromass carbon.** Contents of bacterial (**a**), fungal (**b**), and total (**c**) necromass carbon (C) under warming and control conditions. Total necromass C is calculated as the sum of bacterial and fungal necromass C. **d** Difference in the proportion of total necromass C to soil organic C between the warming and control treatments. Box represents the interquartile range, with blue and yellow indicating control and warming, respectively. Horizontal line and circle within the box show the median and mean value, respectively. The whisker denotes SD ($n = 10$, independent samples). Paired samples $t$-tests (two-sided) were used to determine warming effects on means of microbial necromass C. Source data are provided as a Source Data file.

contents of GluN, MurA, and total amino sugars as the sum of three individual ones were significantly higher under warming ($P < 0.05$; Supplementary Fig. 7). We then used amino sugars to calculate microbial necromass C, and found that warming increased the contents of bacterial, fungal and total necromass C ($P < 0.05$; Fig. 4a–c). With unaltered SOC content between warming and control conditions (Supplementary Table 2), the proportion of microbial necromass C to SOC significantly increased from $28 \pm 1\%$ to $31 \pm 1\%$ under warming ($P < 0.01$; Fig. 4d), reflecting larger contribution of microbial-derived compounds to soil C pool within warming context.

Both $R_h$ (characterizing soil C release) and MAOC (indicating soil C stabilization) were altered by experimental warming. Compared with the control, the warming treatment significantly enhanced $R_h$ by $41 \pm 4\%$ ($P < 0.001$; Fig. 5a), as reported in the first three years of

warming when $R_h$ was elevated by ~59%[31]. For soil C stabilization, the proportion of MAOC under warming was significantly higher than that under control (warming vs. control: $71 \pm 2\%$ vs. $65 \pm 3\%$; $P < 0.05$; Fig. 5b). In contrast to MAOC, the proportion of POC (particulate organic C) showed no significant difference between warming and control (Supplementary Fig. 8a), while heavy POC declined under warming ($P < 0.01$; Supplementary Fig. 8b). These results collectively indicated enhanced soil C loss by microbial respiration but meanwhile stable soil C accumulation derived partly from microbial necromass under warming condition.

## Discussion
Knowledge about consequences of climate warming for soil microbial communities and the C cycling processes they govern is important for

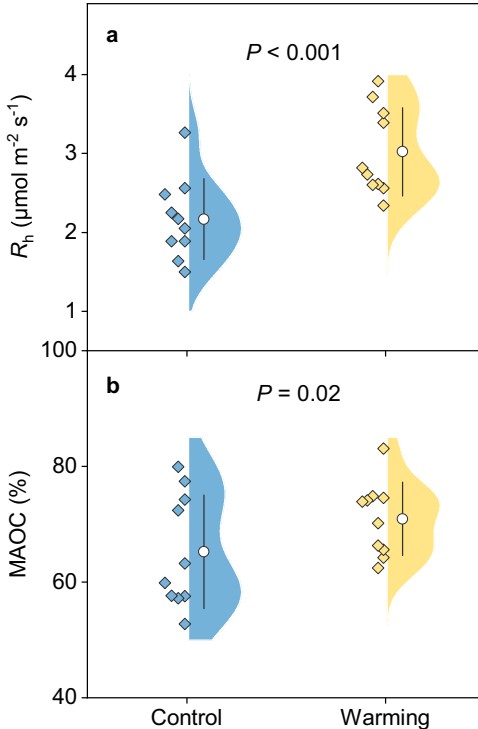

**Fig. 5 | Warming effects on soil carbon release and stability. a** Rate of soil heterotrophic respiration ($R_h$) during growing season in 2020. **b** Proportion of mineral-associated organic carbon (MAOC) to total soil organic carbon in topsoil. Dots on the left of the violin represent the corresponding data from control (blue) or warming (yellow) condition. White circle within the violin shows the mean value, and the error bar denotes SD ($n = 10$, independent samples). Paired samples *t*-tests (two-sided) were used to compare means of $R_h$ and MAOC between the warming and control treatments. Source data are provided as a Source Data file.

understanding and projecting soil C dynamics in permafrost region. Based on a more than half-decade of OTC warming experiment in a permafrost ecosystem, this study provides thorough evidence about microbial responses to climate warming, and explores microbial roles in affecting soil C release and C stabilization. Our results revealed that multiple microbial attributes, including microbial physiology, network complexity, metabolic capacities, and necromass jointly shaped the dual roles of microorganisms in affecting soil C processes under warming scenario.

Our results illustrated that the warming treatment significantly enhanced $R_h$ without detectable changes in microbial community composition (Fig. 5a and Supplementary Table 3), partly invalidating our first hypothesis and indicating other microbial mechanisms underlying $R_h$. On one hand, the increased $R_h$ under experimental warming could be ascribed to the reduction in microbial CUE (Fig. 3c). Higher temperature is generally expected to increase energy cost of maintaining existing biomass, thereby decreasing microbial efficiency[32]. In consistent with this point, our results provide explicit evidence for microbial physiological adjustment towards greater maintenance cost at elevated temperatures in the climate-sensitive permafrost region, without acclimation (i.e., return towards pre-warmed conditions[29]) after more than a half-decade of warming treatment. Accordingly, microbial respiration rate was accelerated more by warming than growth (Fig. 3a, b), leading to increased soil C release through heterotrophic respiration. On the other hand, experimental warming enhanced the complexity of microbial networks, which may further positively influence ecosystem functions related to microbial decomposition[33–35]. Microbial communities could drive ecosystem functions through a myriad of interactions among

different species[34], forming ecological networks with the flow of energy, materials and information[36,37]. More complex networks were established under warming conditions with distinct microbial community composition (Fig. 1 and Supplementary Table 3), suggesting that microbial species better adapting to higher temperatures and other associated edaphic and vegetation changes (such as increased inorganic nitrogen and plant biomass; Supplementary Table 2) could construct new network structures[38]. Consequently, microbial decomposition could be promoted due to the positive effects of network complexity on microbial functions[33–35]. Collectively, these results revealed enhanced $R_h$ driven by reduced microbial CUE and more complex microbial networks in the context of climate warming. Notably, microbial networks are constructed based on pairwise correlations of ASVs which do not prove direct interactions between microbial taxa, so caution is needed when inferring the associations in these co-occurrence networks[39]. In addition, other abiotic factors affected directly (increased soil temperature) or indirectly (such as increased available nutrient) by experimental warming could also be associated with the stimulated $R_h$[40].

Our results also demonstrated that experimental warming significantly increased the content of microbial necromass C and its proportion to total SOC (Fig. 4), which could further increase the stable soil C pool (i.e., MAOC; Fig. 5b) due to the vital contribution of microbial-derived compounds to soil C sequestration[12]. Microbial-derived C enters soil primarily through the in vivo turnover pathway; that is, soil microorganisms first uptake accessible compounds, and then deposit products of anabolism to soil via cell generation, growth and death[12]. Since this pathway is driven by microbial turnover and could be affected by microbial anabolic capacity, it is previously considered that accelerated microbial turnover and more C allocated to growth would promote accumulation of microbial residues[41,42]. Surprisingly, divergent from this view and our second hypothesis, we found that the warming treatment elevated the content of microbial necromass C without altering microbial growth and turnover rate (Fig. 3a and Supplementary Fig. 6a). This divergence could be due to the fact that the accumulation of microbial necromass C in soil is determined by both its input and output, as well as its association with soil minerals[43]. While unaltered microbial growth and turnover rate indicated little difference in necromass formation, the persistence of necromass C could be promoted under warming due to the following two aspects. First, soil microorganisms were more likely to utilize plant-derived compounds, while newly-formed microbial residues were relatively less decomposed and thus accumulated over time. The deduction was supported by the enriched microbial functional genes involved in plant material (i.e., starch, hemicellulose, cellulose and pectin) degradation (Fig. 2b and Supplementary Fig. 5). Moreover, the relative abundance of most functional genes that were enriched under warming was positively associated with root biomass (Supplementary Fig. 9), indicating that with enhanced plant C inputs, experimental warming has the potential to stimulate microbial breakdown of these compounds. Second, warmer conditions could promote the preservation of microbial necromass. In permafrost region where cold climate dominated, elevated soil temperature would favor organo-mineral adsorption reactions like ligand exchange[44,45], such that more microbial-derived C could be stabilized by their associations with soil minerals. Accordingly, the proportion of MAOC that is primarily composed of microbial-derived C significantly increased (Fig. 5b), suggesting enhanced soil C stability under warming condition. Nevertheless, considering that the increase in necromass C is a consequence of long-term accrual, experiments with time-series microbial data are encouraged to better understand the alterations in this stable soil C pool.

Although this study provided important insights into microbial roles in mediating soil C dynamics in response to experimental warming, some limitations still exist. First, the analyses of microbial

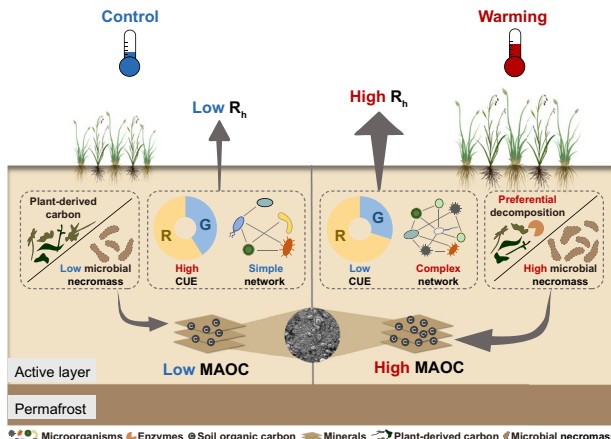

**Fig. 6 | Conceptual diagram showing dual microbial roles in mediating soil carbon dynamics under experimental warming.** The decreased microbial carbon use efficiency (CUE) and more complex microbial network promote soil hetero-trophic respiration ($R_h$), while preferential microbial decomposition of plant-derived carbon leads to accumulation of microbial necromass and thus elevates the proportion of mineral-associated organic carbon (MAOC). R microbial respiration, G microbial growth.

community composition and functional capacities were based on DNA sequencing that provided relative abundance data rather than absolute values. Extending studies adopting quantitative polymerase chain reaction (PCR), spike-ins of microbial cells[46], and enzyme assays should be conducted to enable the quantification of taxonomic and functional genes. Second, the results reported here were derived from one site on the north-eastern Tibetan Plateau. Future research involving more study sites such as coordinated distributed experiments[40,47] is required to examine whether the microbial mechanisms underlying soil C processes observed in this grassland ecosystem are applicable to other permafrost regions.

In summary, based on a more than half-decade of in situ warming experiment in the Tibetan alpine permafrost region, our results revealed the dual roles of microbial communities in mediating two soil C processes (that is, soil C release and stabilization) in response to experimental warming (Fig. 6). Decreased microbial CUE and more complex prokaryotic and fungal networks under experimental warming enhanced $R_h$, whereas preferential microbial decomposition of plant-derived substrates resulted in the accumulation of microbial necromass and hence increased the proportion of stable soil C pool (i.e., MAOC). These findings have several important implications for projecting soil C fate in permafrost regions under future climate warming scenarios. First, given that the accelerated microbial decomposition under experimental warming was partly driven by the decreased CUE, the effect might dampen over time because microbial biomass declined with less C being allocated to growth (Supplementary Fig. 6b, c). Second, the preferential preservation of microbial-derived compounds which are easily associated with soil minerals highlights the importance of microbial anabolism in soil C stabilization[12] under experimental warming. The resultant larger proportion of MAOC, together with the dampened response of microbial respiration, indicates that the future permafrost C-climate feedback might not be as strong as previously thought. Finally, considering that different attributes of microbial communities (such as microbial co-occurrence patterns, metabolic capacities and physiology) affected warming-induced soil C processes, explicitly representing these microbial properties in Earth system models is highly necessary to more accurately predict the fate of permafrost C under changing environment.

## Methods

### Site description and experimental design

The in situ experimental warming site was located at the foot of Wayan Mountain (37°45′N, 100°05′E; 3800 m above sea level) in Gangca County, Qinghai Province, China. This study site is underlain by dis-continuous permafrost, with an active layer thickness of ~4 m. The site is characterized as a swamp meadow ecosystem with mean annual air temperature of −3.4 °C and mean annual precipitation of 466 mm[28,48]. Vegetation at the study site is dominated by sedges mainly including *Kobresia tibetica* and *Carex atrofusca* (accounting for ~75% of the total aboveground biomass)[31]. The soil type is Cambisols according to the FAO system (http://www.fao.org/), with 5% clay, 49% silt, and 46% sand[28,48]. The field warming experiment followed a paired design beginning in June 2013, when ten 4 × 4 m blocks were randomly established in a 50 × 50 m fenced area[28] (Supplementary Fig. 10). Within each block, we diagonally arranged the control and warming plots, with the former being located at one corner (0.8 × 1.2 m) and the latter at the opposite corner with a hexagonal OTC. The OTCs were constructed from transparent polymethyl methacrylate, and were used to warm air and soil throughout the year in the field. Soil temperature and moisture at 5 cm depth were monitored at 30-min intervals with ECH₂O sensors connected to EM50 data loggers (Deca-gon Devices Inc., Pullman, WA, USA). OTCs continually increased topsoil temperature by 0.8–2.2 °C in the warming plots but did not alter soil moisture since 2014 (Supplementary Tables 1 and 2).

After 6-year continual warming treatment, soil sampling was performed in July of 2019 and 2020, and the warming effects on biotic and abiotic factors were examined by comparing the attributes of samples collected from the warming and control treatments at the same year. Briefly, considering that OTC devices primarily warm surface soils[31], three randomized soil cores at 0–10 cm depth were collected from each plot and mixed as a composite. In total, 20 soil samples were acquired (10 from control plots and 10 from warming plots) each year, which were transferred to the laboratory immediately in a cooler. We picked out all roots in each soil sample and sieved the soils through a 2-mm mesh for the subsequent analyses. All roots were rinsed free of attached soil, after which live roots were distinguished from dead roots according to their color and tenacity[28]. Live roots were then oven-dried to a constant mass, and weighed to determine root biomass in 0–10 cm soils to reflect potential changes in plant C input into soils and the associated effects on topsoil microbial function.

To explore warming effects on soil C dynamics, we determined in situ $R_h$ and SOC distribution in different soil fractions, and also measured a suite of soil physicochemical properties. To examine the roles of soil microorganisms in affecting soil C processes, we deter-mined microbial community composition and microbial necromass C for soil samples collected in 2019 (also used for measurements of C distribution and soil physicochemical properties). To further provide thorough evidence for microbial responses to experimental warming, microbial functional and physiological properties were determined for soil samples obtained in 2020. Each attribute was compared between the warming and control treatments for the samples collected at the same year to reflect warming effects.

### Heterotrophic respiration measurements and soil chemical analyses

In the field, we measured $R_h$ using a trenching approach (i.e., root exclusion technique)[49]. We focused on $R_h$ because it was a con-sequence of microbial activities. Briefly, a PVC collar (20 cm diameter, 62 cm high) was inserted into the soil to a depth of 60 cm in June 2013 to sever existing roots and inhibit future root growth. The collar could eliminate the majority of roots since the top 60 cm soil contained more than 96% of roots at our study site[31]. Soil respiration measured from

this "root-free" collar was considered as $R_h$. At least 1 day prior to each measurement, living plants at the soil surface were removed manually from the collar when necessary[50], avoiding disturbance to soil profile and inhabitants. Nevertheless, despite a widely used method for $R_h$ measurement[51–53], the exclusion of roots would reduce root exudates and thus underestimate $R_h$ due to the lack of rhizosphere priming effect[49]. $R_h$ was determined using a LI-8100A automated soil $CO_2$ flux system (Li-Cor Inc., Lincoln, NE, USA) between 9:00 a.m. and 12:00 p.m. (local time) twice to three times per month from May to October in 2020. Meanwhile, we determined the NDVI using a multi-spectral camera (ADC Lite, Tetracam, Chatsworth, CA, USA) to reflect warming effects on the growth of above-ground vegetation.

A suite of chemical properties was measured for the soil samples to examine their potential effects on microbial communities under experimental warming. Briefly, total C content for soil samples was measured by an element analyzer (Multi EA 4000, Analytik Jena, Jena, Germany), which was treated as organic C because no inorganic C was detected in the samples[31]. DOC was extracted using 0.5 M $K_2SO_4$ and determined by a Multi N/C 3100 TOC/TN$_b$ analyzer (Analytik Jena, Jena, Germany). The $NH_4^+$-N and $NO_3^-$-N concentrations were measured with a continuous flow analyzer (SEAL AutoAnalyzer 3, SEAL Analytical, Norderstedt, Germany) after extracting soil samples with 1 M KCl solution. Soil pH was measured in a 1:2.5 soil-to-water mixture by a pH meter (PB-10, Sartorius, Germany).

## Soil organic matter fractionation
To explore warming effects on soil C distribution, we adopted a density and size-combined fractionation approach to separate bulk soil organic matter into three fractions[54]. Specifically, 10 g air-dried soil samples were suspended in 50 ml of 1.6 g cm$^{-3}$ NaI solution. The suspension was shaken by hand to ensure complete soil wetting and then dispersed by ultrasound (250 J ml$^{-1}$) to release occluded particulate organic matter (POM). Thereafter, the suspension was allowed to stand for 30 min and centrifuged for 30 min to ensure sedimentation of the heavy fraction. The light fraction (i.e., POM with density <1.6 g cm$^{-3}$) was collected and rinsed thoroughly of NaI with deionized water on the GF/F filter (Whatman GmbH, Dassel, Germany). The remaining soil material (>1.6 g cm$^{-3}$) was rinsed clean of NaI and further sieved through a 53 μm screen to separate heavy POM (>53 μm) and mineral-associated organic matter (MAOM; <53 μm). All fractions (POM, heavy POM and MAOM) were oven-dried, weighed and analyzed for C content using an element analyzer (Multi EA 4000, Analytik Jena, Jena, Germany), after which C distribution in each fraction was calculated (termed as POC, heavy POC and MAOC, respectively). We recovered on average >93% of the initial soil masses and >97% of soil C across all samples involved in this study.

## DNA extraction, amplicon sequencing and bioinformatic analyses
We conducted amplicon sequencing to assess prokaryotic and fungal community composition under warming and control conditions. Specifically, 0.5 g soils stored at −80 °C were used for DNA extraction (FastDNA SPIN Kit for Soil, MP Biomedicals). DNA concentration and quality were measured by NanoDrop One Spectrophotometer (Thermo Fisher Scientific, Waltham, MA, USA). Thereafter, primer sets, 515F (5′-GTGCCAGCMGCCGCGGTAA-3′) and 806R (5′-GGACTACH VGGGTWTCTAAT-3′) targeting the V4 hypervariable region of prokaryotic 16S rRNA gene[55], and ITS3 (5′-GCATCGATGAAGAACGCAGC-3′) and ITS4 (5′-TCCTCCGCTTATTGATATGC-3′) for fungal ITS2 region[56], were used for amplification. The PCR amplification for 16S rRNA gene was performed at 94 °C for 5 min, followed by 30 cycles of 94 °C for 30 s, 53 °C for 30 s, and 72 °C for 30 s, with a final extension at 72 °C for 8 min. For ITS2 region, PCR was conducted under the following conditions: 98 °C for 30 s, followed by 32 cycles of 98 °C for 10 s, 56 °C for

20 s, and 72 °C for 30 s, with a final extension at 72 °C for 8 min. PCR products were detected by 1% agarose gel electrophoresis and purified using the EZNA Gel Extraction Kit (Omega, Norcross, GA, USA). Purified PCR products were then pooled to libraries and sequenced on the Illumina Hiseq 2500 platform (Illumina, San Diego, CA, USA) with 2 × 250 base pair kits (Guangdong Magigene Biotechnology Co., Ltd. Guangzhou, China).

The raw reads were trimmed by TrimGalore (v0.6.10) (https://github.com/FelixKrueger/TrimGalore) to remove low-quality reads (quality score <25, <100 bp), and then merged using USEARCH[57] (v11). The primers and low-quality reads (<200 bp) were further removed using cutadapt plugin within QIIME2[58] (v2024.2) and USEARCH, respectively. Thereafter, clean reads were dereplicated using the command -fastx_uniques and then clustered into ASVs using the unoise3 algorithm[59] with USEARCH. The selected ASV representative sequences for 16S and ITS were annotated taxonomically with Silva[60] (v138) and Unite[61] (v9.0) databases respectively, using the SINTAX algorithm[62] in USEARCH. ASVs that were not classified as prokaryotes (i.e., bacteria and archaea) or fungi were removed. The ASV matrices for 16S and ITS were rarefied to their minimum sequence number across all samples respectively for the subsequent analyses (i.e., calculation of diversity and community composition, and network construction).

## Metagenomic sequencing and data processing
We performed metagenomic sequencing on 20 soil samples to explore warming effects on potential microbial functions. Metagenomic libraries of the samples were constructed and sequenced on the Illumina HiSeq 2500 platform (Illumina, San Diego, CA, USA) with a 2 × 150 base pair kits (Guangdong Magigene Biotechnology Co., Ltd. Guangzhou, China). We acquired 1.5 billion raw reads in total from the 20 samples. For sequence processing, Trimmomatic[63] (v0.39) was used to remove adapters and trim raw reads. The filtered clean data of each sample were firstly assembled using MEGAHIT[64] (v1.2.9) with parameters: k-min 35, k-max 95, and k-step 20, and the scaftigs shorter than 500 bp were excluded. Then the reads that were unmappable to the scaftigs were pooled together and further co-assembled using the same parameters as those for individual samples. The resulting assemblies from the two steps were combined, and used for prediction of open reading frames (ORFs) by prodigal[65] (v2.6.3). Subsequently, ORFs were clustered and dereplicated with 95% sequence identity using linclust algorithm from MMseqs2[66] to acquire non-redundant gene catalog. To uncover changes in the potential microbial metabolic capabilities with warming, the predicted genes were annotated against different databases. Annotation against the CAZy[67] (CAZyDB.07312018) was performed using run_dbcan[68]. Substrate utilization of enzymes from the CAZy family was acquired based on CAZy database and previous literatures (Supplementary Table 5). In addition, genes were annotated via DIAMOND[69] (v0.9.14) to the KEGG database[70] (downloaded 1-April-2020; https://www.kegg.jp/kegg/download/). KEGG Orthology (KO) terms involved in the pathways of substrate utilization were acquired from the KEGG database and published literature[71]. Clean reads were mapped to gene catalog using BWA[72] (v0.7.17), and the relative abundance of each gene for sample S was calculated as follows[73]:

$$RA_i = \frac{r_i/L_i}{\sum_j \frac{r_j}{L_j}} \tag{1}$$

where $RA_i$ is the relative abundance of gene $i$ in sample $S$, $r_i$ is the counts that gene $i$ can be detected in sample $S$, and $L_i$ is the length of gene $i$.

## Determination of microbial physiology

To evaluate responses of microbial physiology to experimental warming, we determined microbial growth rates and CUE by a substrate-independent method according to the incorporation of $^{18}O$ into microbial DNA[29,74]. Specifically, fresh soil samples previously stored at 4 °C were adjusted to 55% water holding capacity (WHC) and preincubated at 15 °C (the temperature soils can reach during growing season) for 7 days. Then two replicates (one as labeled sample and the other as natural abundance sample) of 300 mg preincubated soils were weighed into 2 ml reaction tubes, and each open tube was placed into a 50 ml vial sealed with a crimp cap. The vials with soil samples as well as empty ones as control were flushed with $CO_2$-free air. Thereafter, labeled samples were added with $^{18}O$-$H_2O$ to reach 70% of WHC and 20 at% of $^{18}O$ in the final soil water, while the natural abundance sample received the same volume of non-labeled molecular biology grade water. The vials were incubated at 15 °C for 24 h, after which headspace $CO_2$ concentration was determined by an Agilent 7890A gas chromatograph (Agilent Technologies, Santa Clara, CA, USA). Microbial respiration rate was calculated based on the difference in $CO_2$ concentration between samples and empty controls.

Immediately after gas sampling, the reaction tube was closed and put into liquid $N_2$. Subsequently, DNA of soil samples was extracted with FastDNA SPIN Kit for Soil (MP Biomedicals) following the instruction manual with two modifications[74]: (1) the time of first centrifugation was extended to 15 min; (2) the entire binding matrix suspension was transferred onto the filter. The weight of the DNA extract was recorded and the DNA concentration was then quantified using the Quant-iT PicoGreen dsDNA Assay Kit (Thermo Fisher). $^{18}O$ abundance and total O content were analyzed using an elemental analyzer coupled to a 253 plus isotope ratio mass spectrometer via a Conflo IV (Thermo Fisher Scientific, Waltham, MA, USA). The amount of DNA produced during incubation time ($DNA_{produced}$; μg) was calculated as follows[29]:

$$DNA_{produced} = O_{total} \times \frac{O_{excess}}{100} \times \frac{100}{O_{lable}} \times \frac{100}{31.21} \quad (2)$$

where $O_{total}$ represents the total O content of the dried DNA extract (μg), $O_{excess}$ is the surplus $^{18}O$ abundance (at%) of labeled samples compared to the mean at% $^{18}O$ of natural abundance samples, $O_{lable}$ is the at% $^{18}O$ in the final soil solution of the labeled sample, and the constant 31.21 denotes the proportional mass of O (%) in DNA according to an average DNA molecule. To convert the produced DNA to the equivalent production of microbial biomass carbon (MBC), we conducted chloroform fumigation-extraction procedure followed by analyses on a Multi N/C 3100 TOC/TN$_b$ analyzer (Analytik Jena, Jena, Germany) to determine MBC (conversion factor 0.45)[75]. MBC production, i.e., microbial growth ($G$, μg C g$^{-1}$ soil dry mass h$^{-1}$) for each sample was calculated as:

$$G = \frac{\frac{C_{mic}}{DNA_{mic}} \times DNA_{produced}}{w \times t} \quad (3)$$

where $C_{mic}$ and $DNA_{mic}$ are contents of a sample's microbial biomass carbon (μg C g$^{-1}$ soil dry mass) and DNA (μg DNA g$^{-1}$ soil dry mass), $w$ is the dry mass of soils used for incubation (g), and $t$ is the incubation time (h). Given that microbial physiological processes operate per unit of biomass, we also acquired biomass-specific growth ($G_m$; mg C g$^{-1}$ $C_{mic}$ h$^{-1}$) and respiration ($R_m$; mg C g$^{-1}$ $C_{mic}$ h$^{-1}$). Microbial CUE was then determined as follows:

$$CUE = \frac{G}{G + R} \quad (4)$$

where $R$ is microbial respiration (μg C g$^{-1}$ soil dry mass h$^{-1}$). Microbial turnover rate ($T_m$; day$^{-1}$) was estimated as:

$$T_m = \frac{G}{C_{mic}} \times 24 \quad (5)$$

## Analyses of amino sugars

Amino sugars in soil samples were analyzed according to a previous protocol[76] and used to calculate microbial necromass C. Briefly, freeze-dried and ground soil samples containing more than 0.3 mg N were hydrolyzed with 6 M HCl containing 100 μl myo-inositol (internal standard) for 8 h at 105 °C. The hydrolysate was filtered, adjusted to pH 6.6–6.8 and centrifuged. Thereafter, the supernatant solution was evaporated to dryness at 52 °C, and amino sugars were re-dissolved in 5 ml methanol, transferred to vials and dried at 45 °C with $N_2$. The residues were added with 1 ml deionized $H_2O$ and 100 μl standard N-methylglucamine, and lyophilized. Then aldononitrile acetate derivatization for amino sugars was processed by adding a derivatization reagent containing hydroxylamine hydrochloride (32 mg ml$^{-1}$) and 4-(dimethylamino)pyridine (40 mg ml$^{-1}$) in 4:1 (v/v) pyridine-methanol, and heating at 75–80 °C for 35 min. The derivatives were further acetylated with 1 ml acetic anhydride and the solution was reheated for 25 min (75–80 °C). Then, 1.5 ml dichloromethane and 1 ml of 1 M HCl were added successively. Excessive derivatization reagents were reacted with 1 M HCl and removed by washing thrice with deionized water. The organic phase was dried at 45 °C with $N_2$, and resuspended in ethyl acetate-hexane (1:1), which was finally analyzed by an Agilent 7890B gas chromatograph (Agilent Technologies, Santa Clara, CA, USA) equipped with a HP-5 column (30 m length × 0.25 mm diameter × 0.25 μm thickness) and flame ionization detector. The content of individual amino sugar (i.e., GluN, GalN, and MurA) was calculated based on the internal standard, and sum of the three individual amino sugars was used to reflect soil amino sugar pool. Microbial necromass C was calculated according to the content of MurA ($m$) and GluN ($n$) as follows[30]:

$$\text{Bacterial necromass C} = 45 \times m \quad (6)$$

$$\text{Fungal necromass C} = \left(\frac{n}{179.17} - 2 \times \frac{m}{251.23}\right) \times 179.17 \times 9 \quad (7)$$

where 45 and 9 represent the conversion value from MurA to bacterial necromass C, and fungal GluN to fungal necromass C, respectively. 179.17 is the molecular weight of GluN, and 251.23 is the molecular weight of MurA. Total microbial necromass C is the sum of bacterial and fungal necromass C.

## Statistical analyses

We conducted the following analyses to estimate warming effects on different microbial attributes as well as soil C dynamics and environmental factors: first, we performed paired samples $t$-tests to compare the means of each variable between warming and control conditions. The variables included R$_h$, C distribution in soil fractions, plant variables (NDVI and root biomass), soil physicochemical properties (soil temperature and moisture, SOC, DOC, $NH_4^+$-N, $NO_3^-$-N, and pH), as well as microbial physiological properties (microbial growth, respiration, CUE and turnover) and microbial necromass C.

Second, we explored whether and how experimental warming affected microbial diversity, community composition and co-occurrence patterns. The analyses were conducted using R software 4.1.0[77] with the vegan package[78] when not specified. Specifically, we calculated alpha diversity indices including richness and Shannon−Wiener index for prokaryotes and fungi, and compared the

means between control and warming conditions using paired samples *t*-tests. Then we examined the differences in prokaryotic and fungal community composition between the warming and control treatments using three non-parametric multivariate analyses, including permutational multivariate analysis of variance (Adonis), analysis of similarities (ANOSIM), and multi-response permutation procedure (MRPP) based on Bray–Curtis distance.

To further examine warming effects on microbial co-occurrence patterns, we constructed molecular ecological networks for prokaryotes and fungi under control and warming conditions, respectively. Network construction and analyses were all carried out using the integrated network analysis pipeline (iNAP) at https://inap.denglab.org.cn/[79]. Briefly, only prokaryotic and fungal ASVs presented in all the 10 samples were retained for correlation calculation to ensure network reliability. Rarefied sequence data were log-transformed before obtaining Pearson correlation matrix, and the resulting matrix was analyzed by a random matrix theory (RMT) based approach[37] to determine the threshold of the Pearson correlation. Uniform threshold was selected for control and warming for network construction (0.91 for prokaryotes and 0.79 for fungi). Then, network topology characterization, network randomization and module separation were processed with default parameter setting. After module detection, relative modularity was calculated to reflect how modular an empirical network is compared with the mean expected modularity[38]. Keystone nodes were identified according to the topological properties of each node (that is, within-module connectivity ($Z_i$) and among-module connectivity ($P_i$)[80,81]), and module eigengene analysis was conducted to explore modules' response to environmental changes (see Supplementary Note 2 for details). Networks were visualized using Cytoscape 3.9.0[82]. Finally, to assess the differences in networked communities between control and warming, non-parametric multivariate analyses including Adonis, ANOSIM, and MRPP were conducted. Variation partitioning analysis based on redundancy analysis was also performed to discern the contributions of the plant and edaphic variables to the variations in the networked communities.

Third, warming effects on potential microbial functions were examined. Briefly, statistically differentially abundant CAZy gene families were identified through LEfSe method[83] at http://galaxy.biobakery.org/. LDA scores higher than 2.0 with a *P* value less than 0.05 were considered significantly enriched. In addition, paired samples *t*-tests were conducted to determine the differences in relative abundance of total CAZy gene families as well as the sum of KOs involved in the specified substrate utilization.

### Reporting summary

Further information on research design is available in the Nature Portfolio Reporting Summary linked to this article.

## Data availability

All data supporting the findings are available in the Figshare data repository (https://doi.org/10.6084/m9.figshare.25974622.v2)[84] and Supplementary Information. The sequence data generated in this study have been deposited in the NCBI Sequence Read Archive (SRA) under accession number PRJNA1113361. Source data are provided with this paper.

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

## Acknowledgements

We are grateful to Prof. Yunfeng Peng at Institute of Botany, Chinese Academy of Sciences and Dr. Fei Li at Taizhou University for experimental platform establishment. We also appreciate Prof. Tida Ge at Ningbo University for assistance in CUE determination, Yutong Song and Luyao Kang from Yuanhe Yang's group for providing helpful suggestion in data analyses, and Yang Liu and Qinlu Li from Yuanhe Yang's group for assistance in field measurements. This work was supported by the Second Tibetan Plateau Scientific Expedition and Research (STEP) program (2019QZKK0106, Y.Y.), the National Natural Science Foundation of China (31988102 and 32425004, Y.Y., and 32301436, S.Q.), the New Cornerstone Science Foundation through the XPLORER PRIZE (Y.Y.), and the China National Postdoctoral Program for Innovative Talents (BX20220339, S.Q.).

## Author contributions

Y.Y. and S.Q. designed the research. S.Q., D.Z., and B.W. performed field sampling and laboratory experiments. B.W. conducted field measurements. S.Q. analyzed the data. S.Q. and Y.Y. wrote the manuscript.

## Competing interests

The authors declare no competing interests.
