## [Peer Review File · Nature Communications]

Dual roles of microbes in mediating soil carbon dynamics in response to warmingREVIEWER COMMENTS

Reviewer #1 (Remarks to the Author):

The manuscript is in general very clear, detailed, and coherent. The research topic is an important one and the authors have applied a robust mixture of methodologies (CUE, necromass, genomics, geochemistry, etc) to examine their hypotheses.

Noteworthy results:

-decreased CUE

-increased network connections, necromass, and necromass contribution to SOC

Significance etc: Important study looking at the in/stability of active layer soil carbon and processes and taxa affecting it. This study used a novel combination of techniques and scientific concepts to build on previous work in the Qinghai-Tibet Plateau experimental warming station and other global works examining soil carbon stability. It will be relevant to microbial ecologists, climate modellers, and biogeochemists.

Issues:

There are questions about the methods used to obtain the network, community, and Rh findings, as well as questions about soil warming and the timeframe of the study.

Concerns potentially affecting study findings:

- I expected after reading the abstract/ introduction that the analysis would be between soils or conditions comparing 2013/14 to 2019/20. It took a while to realise that this study compares 2019 to 2020 and is therefore a study of one year of warming not six years. Please be very clear about what is being compared in this study. Further to this, previous reports from this research site document soil temperatures several degrees higher than those reported here. Given that the comparison is made between soils one year apart not six years apart, and the lower soil temperatures, it is hard to justify the statements that the soils investigated have been warmed for six years. Please clarify this and preferably provide temperature information for the six years as it is critical to the understanding and context of the study.

-Community analyses (e.g. fig S1e,f , Table S2), were done at phyla level to conclude no change. Phyla level is not a relevant level for comparisons of community. It would be more meaningful if no change was found at family, genus or even species level. Given that species level assignment is not generally possible with short amplicons then it would be of interest to the readers to see analyses from phyla down to genus level to see if/where there is taxonomic difference.

-Ten samples is low statistical power for network construction, can the use of this low number of samples be justified with citations?

-Were gene relative abundances adjusted for gene length? This can affect the likelihood of

documenting a gene being seen, as longer genes have increased probability of being detected than shorter genes.

-This study presents a novel way to remove autotrophic respiration from soil respiration measurements as compared to methods that block sunlight from the chamber ie cloth cover. I am concerned that the continual removal of plants throughout the growing season would have disturbed the soil profile, altered root exudate profile, disturbed the soil inhabitants (potentially selecting for those most able to handle the disturbed conditions), and altered soil inhabitant function. Can the authors provide evidence of a lack of disturbance? If I misunderstand the method then please describe more clearly so others don't make the same mistake. There is also the possibility that mosses, lichens, and free-living cyanobacteria were present and emitting (autotrophic) CO₂ during measurements, can the authors show evidence that this was not the case?

- What is the justification for removal of Archaea from the 16S gene dataset. Was the metagenomic data cleaned of archaeal/eukaryote non-fungal/viral sequences to only contain bacterial and fungal reads? Was ORF calling done in such a way as to only include bacterial and fungal sequences?

-Ding et al 2017 <https://doi.org/10.1038/ngeo2945>, documents carbon accumulation in the active layers of the QTP, a finding which presumedly led to this follow up work to determine processes connected to this accumulation. Please cite this important work and provide more context for the readers and explain why SOC increase was not found in this study.

-I did not see any section that acknowledges issues or limitations of the methods or design. DNA sequences and their relative abundances do not give absolute values. Absolute values can be obtained by doing quantitative PCR and enzyme assays, or maybe could be linked to the DNA production values in a novel calculation. This project is only one study site, there is therefore no replication other than technical replicas so care needs be taken with extrapolations.

Concerns affecting reader interpretation:

-it would be nice to see photos of the field site and the passive warming chambers in-situ

-boxplots are easier for readers to interpret paired comparisons of means rather than bar plots

-were the same thermal cycle conditions used for both the 16S and ITS reactions?

-It is currently more accepted to analyse rRNA gene amplicons using ASVs, and less acceptable to use clustering methods. It would be worthwhile to cite a paper that shows clustering methods, ie OTUs, are as robust a method as ASVs.

-please consider dividing the number of keystone nodes per network by the number of nodes as the size of the network may bias these results and its more meaningful to see if there are changes in the ratio of keystone to non-keystone OTUs

-edaphic and plant variables could be included in the networks to see which modules they were connected to

-no RNA analysis was done so you could not have transcripts per million

-were the RMT thresholds applied to Pearsons correlation or something else? Please clarify

-can the networks be plotted with correlation weighted connector lines? Please note that the lines do not represent interactions, they represent correlation which does not prove interaction

-it is unclear what value plotting networks with module colours adds, is there a more meaningful colour scheme you can use?

-regarding references for node classification: Olesen et al is absolutely a good reference but Guimerà et al is a better second, <https://www.ncbi.nlm.nih.gov/pmc/articles/PMC2175124/> the other references are more about interpretation and so could be removed.

-while the node topological classification scheme is respected and well documented, it does look somewhat subjective in Fig 1e,f. Can further justification be given on these choices? Or other choices made?

- for all the sequence analyses (whether marker genes or functional genes) all abundances are relative, please fix throughout the manuscript

-it is unclear what is meant by “functional structure“ in the CAZy section

-the relationship between MBC and CUE was calculated with all data points, it of interest to see if this ratio is different between control and warming (given the caveat of low statistical power with so few data points)

-Current best practice with molecular microbiology is to deposit amplicon and metagenome sequences in a public repository/database and supply the accession numbers in the manuscript along with a (private pre-publication) link for the editor/reviewers to verify

-It is also current best practice to deposit scripts used for analysis in a public repository with its own citable DOI eg figshare, zenodo, github

Reviewer #2 (Remarks to the Author):

This manuscript titled “Dual roles of microbes in mediating soil carbon dynamics in response to warming” explores how microbes influence carbon cycle dynamics under experimental warming in permafrost on the Tibetan Plateau using a powerful combination of ecosystem scale and microbial

community scale measurements as well as genomic inference. This study helps fill a knowledge gap in permafrost ecology, providing a view into the contrasting ways that microbes impact carbon release and storage. These results are important for future modeling efforts and meta-analyses to understand how climate warming impacts soil microbial carbon cycling.

The authors showed that while microbial community composition was unchanged after 6 years of warming using OTCs, the presence of functional genes indicated increased potential for plant polysaccharide degradation. Microbial community growth and turnover was unchanged, yet heterotrophic respiration, mineral associated carbon, and microbial necromass all increased with warming. These results indicated that microbes are both responsible for the release of carbon through respiration and the stabilization of organic carbon through necromass C production. The suite of tools used in this study and the measurements made were well documented and appropriate for their hypothesis testing.

A significant result from this study is that under warming, microbes contributed to soil carbon loss, yet helped stabilize the soil carbon pool by contributing necromass C. However, the authors should clarify how to reconcile unchanged turnover with an increase in necromass C with warming. Increased necromass is controlled by increased microbial growth and turnover (Sokol, N. et al. "Life and death in the soil microbiome: how ecological processes influence biogeochemistry." *Nature Reviews Microbiology* 20.7 (2022): 415-430.) The authors acknowledge this on lines 246-7, that the microbes may be preferentially degrading plant derived compounds, but this fails to explain how necromass C may have increased with unchanged turnover. Please add further interpretation to the discussion. Alternatively, this could be a technical problem since necromass samples were collected in 2019 and growth rate was measured in samples from 2020. Can you measure necromass C on the 2020 samples?

specific points:

Lines 32-34: add "likely" before "...due to the preferential microbial metabolism..." This study doesn't provide direct evidence that necromass comes from plant derived compounds.

Line 156: add something like "potential for" prior to "...enhanced utilization..."

Lines 253-4: Gene presence is only functional potential. Qualify the statement "...indicating that enhanced plant C input by experimental warming would stimulate microbial breakdown of these compounds." With "...experimental warming has the potential to stimulate microbial breakdown..."

Line: OTUs tend to overestimate bacterial enrichment, unless rarefaction and 99% id clustering (Chiarello, M. et al. Ranking the biases: The choice of OTUs vs. ASVs in 16S rRNA amplicon data analysis has stronger effects on diversity measures than rarefaction and OTU identity threshold." *PLoS One* 17.2 (2022): e0264443.). Consider ASV generation via a denoising technique.

Line 404: The Silva 138 database was released in 2019 and has important taxonomy updates. This version should be used instead of 132.

Line 433: TPM = transcripts per million, but your study didn't analyze gene transcripts.

Responses to Reviewer #1

[Comment 1] The manuscript is in general very clear, detailed, and coherent. The research topic is an important one and the authors have applied a robust mixture of methodologies (CUE, necromass, genomics, geochemistry, etc) to examine their hypotheses. Noteworthy results: decreased CUE; increased network connections, necromass, and necromass contribution to SOC. Significance etc: Important study looking at the in/stability of active layer soil carbon and processes and taxa affecting it. This study used a novel combination of techniques and scientific concepts to build on previous work in the Qinghai-Tibet Plateau experimental warming station and other global works examining soil carbon stability. It will be relevant to microbial ecologists, climate modellers, and biogeochemists. Issues: There are questions about the methods used to obtain the network, community, and Rh findings, as well as questions about soil warming and the timeframe of the study.

[Response] Thanks for the reviewer's positive comments! We have clarified the issues that the reviewer pointed out and made revisions in the manuscript, which greatly improved our paper. Please see our responses to the comments listed below for detailed modifications.

Concerns potentially affecting study findings:

[Comment 2] I expected after reading the abstract/ introduction that the analysis would be between soils or conditions comparing 2013/14 to 2019/20. It took a while to realise that this study compares 2019 to 2020 and is therefore a study of one year of warming not six years. Please be very clear about what is being compared in this study. Further to this, previous reports from this research site document soil temperatures several degrees higher than those reported here. Given that the comparison is made between soils one year apart not six years apart, and the lower soil temperatures, it is hard to justify the statements that the soils investigated have been warmed for six years. Please clarify this and preferably provide temperature information for the six years as it is critical to the understanding and context of the study.

[Response] Sorry for the confusion! First we would like to mention that the *in situ* warming experiment was established in 2013, and the OTC devices were placed in the field year-round to warm soils until sampling in 2019/2020. Thus, the soils have been continually warmed for six years until sampling in 2019, and for seven years in 2020. In this study, the samples collected in 2019 were used to determine microbial

community composition and necromass C. To further provide thorough evidence for microbial responses to experimental warming, microbial functional and physiological properties were determined for soil samples obtained in 2020. For all the measured attributes, we compared warming treatment to control, and the samples being compared were collected from the same year. That is, we compared neither samples of 2019 to 2020, nor 2013/14 to 2019/20. To avoid the potential confusion, we have clearly stated what is being compared in *Abstract/Introduction* and also other parts of the manuscript (Page 2, lines 30-31; Page 5, lines 90-92; Page 17, lines 350-353).

Following the reviewer’s suggestion, we have also provided the data of soil temperature in control and warming plots each year from 2014 to 2020 in the *Supplementary information* (Table S1 and Table R1), and described the warming effects on soil temperature in the revised MS (Page 6, line 112-113). As shown in Table R1, OTC devices significantly increased soil temperature by 0.8-2.2°C across years (all $P < 0.05$), which was comparable to the range of temperature increase in other OTC warming experiments (Bokhorst *et al.*, 2013). In addition, as mentioned by the reviewer, soil temperature in both control and warming plot was lower in recent years than the first years, which could be due to the shading effects of litter after fencing (Facelli & Pickett, 1991). Nevertheless, considering that we compared warming treatment with control, the alterations in ambient soil temperature would have limited effects on our comparison.

Table R1 Effects of experimental warming on topsoil temperature from 2014 to 2020.

Year	Soil temperature (°C)	
	Control	Warming
2014	8.6 ± 0.13	10.8 ± 0.13**
2015	7.7 ± 0.15	9.3 ± 0.20**
2016	8.4 ± 0.16	9.4 ± 0.26**
2017	7.5 ± 0.15	8.6 ± 0.27**
2018	7.7 ± 0.14	8.5 ± 0.25*
2019	6.5 ± 0.14	7.5 ± 0.22**
2020	6.2 ± 0.14	7.7 ± 0.16**

Data are reported as means \pm standard errors ($n = 10$). Soil temperature is the mean value in growing season (May to October) measured by ECH₂O sensors at 5 cm depth. * $P < 0.05$, ** $P < 0.01$ according to paired samples t-tests (two-sided).

[Comment 3] Community analyses (e.g. fig S1e,f, Table S2), were done at phyla level to conclude no change. Phyla level is not a relevant level for comparisons of community. It would be more meaningful if no change was found at family, genus or even species level. Given that species level assignment is not generally possible with short amplicons then it would be of interest to the readers to see analyses from phyla down to genus level to see if/where there is taxonomic difference.

[Response] Good comments! Following the reviewer’s suggestion, **we have conducted linear discriminant analysis (LDA) effect size (LEfSe) analysis to examine taxonomic difference of prokaryotes and fungi from phyla down to genera level** (Fig. R1). According to the results of LEfSe, differentially abundant prokaryotic taxa were detected at order, family and genus levels (Fig. R1a) and differentially abundant fungal taxa were found from phyla to genera levels (Fig. R1b). We have added the results (Fig. S2) and described these differences in detail in the *Supplementary information* (Pages 2-3, lines 22-51).

Fig. R1 Histogram of the linear discriminant analysis (LDA) scores computed for topsoil prokaryotic (a) and fungal (b) taxa differentially abundant (LDA score > 2, $P < 0.05$) between warming and control treatment. p_, phylum; c_, class; o_, order; f_, family; g_, genus.

[Comment 4] Ten samples is low statistical power for network construction, can the use of this low number of samples be justified with citations?

[Response] We acknowledge that the sample size (ten) might be small for network construction, which was due to the low replicates of *in situ* manipulative experiments constrained by experimental cost and workload. Nevertheless, **previous studies that constructed networks based on *in situ* manipulative experiments always had low sample size**, such as 6 or 8 replicates under experimental warming (Cheng *et al.*, 2021 *Molecular Ecology*; Zhou *et al.*, 2021 *Global Change Biology*), 12 replicates under elevated CO₂ (Zhou *et al.*, 2011 *mBio*), and 10 replicates under land use change (Khan *et al.*, 2019 *FEMS Microbiology Ecology*). **To ensure network reliability as possible, we only retained ASVs presented in all the samples for network construction** (Cheng *et al.*, 2021). We have mentioned this point in the revised MS (Page 28, lines 588-589). Thanks for your understanding!

[Comment 5] Were gene relative abundances adjusted for gene length? This can affect the likelihood of documenting a gene being seen, as longer genes have increased probability of being detected than shorter genes.

[Response] Yes, gene relative abundances were adjusted for gene length using the following equation (Qin *et al.*, 2012 *Nature*):

$$RA_i = \frac{r_i/L_i}{\sum_j \frac{r_j}{L_j}}$$

where RA_i is the relative abundance of gene i in sample S , r_i is the counts that gene i can be detected in sample S , and **L_i is the length of gene i** . We have provided the equation in the revised MS to make this information clearer (Pages 22-23, lines 477-481).

[Comment 6] This study presents a novel way to remove autotrophic respiration from soil respiration measurements as compared to methods that block sunlight from the chamber ie cloth cover. I am concerned that the continual removal of plants throughout

the growing season would have disturbed the soil profile, altered root exudate profile, disturbed the soil inhabitants (potentially selecting for those most able to handle the disturbed conditions), and altered soil inhabitant function. Can the authors provide evidence of a lack of disturbance? If I misunderstand the method then please describe more clearly so others don't make the same mistake. There is also the possibility that mosses, lichens, and free-living cyanobacteria were present and emitting (autotrophic) CO₂ during measurements, can the authors show evidence that this was not the case?

[Response] Sorry for the confusion about R_h measurement! First we would like to introduce the method (*i.e.*, root exclusion technique) briefly. By inserting a deep collar into soil and **removing the above-ground plants** one year prior to measurement, roots within the collar died and the soils were considered root-free to measure R_h (Dorrepaal *et al.*, 2009; Kuzyakov, 2006). During following measurements, we only clipped the living plants **at the soil surface** when necessary, and thus **the disturbance to soil profile and inhabitants could be negligible** as shown in Fig. R2. In addition, as shown in Fig. R2 (and also in other plots according to our observation), there was **no obvious mosses, lichens, and cyanobacteria within the collars**, and thus they would contribute little to CO₂ emission. To avoid the potential confusion, we have added more details about the above-mentioned points in the revised MS (Page 18, lines 378, 381-384).

As for the root exudate, the exclusion of root would indeed reduce exudate and the rhizosphere priming effect (Kuzyakov, 2006). **We have mentioned this limitation of the method in the revised MS** (Page 18, lines 384-387). Nevertheless, despite the potential limitation, the root exclusion technique has been widely used by previous studies (Dorrepaal *et al.*, 2009 *Nature*; Hasselquist *et al.*, 2012 *Global Change Biology*; Nottingham *et al.*, 2020 *Nature*), and all the plots have undergone the same treatment, which would have limited effect on the comparison between warming and control treatment. Thanks for your understanding!

Fig. R2 Photo for a warming plot showing the collars for soil respiration measurements. In the plot, a deep PVC collar (20 cm diameter, 62 cm high) was used for heterotrophic respiration (R_h) measurement, and a shallow PVC collar (20 cm diameter, 5 cm high) was used for the measurement of total soil respiration (R_s). Here we mainly focused on R_h which was driven by microbial activities. Photo was taken by Bin Wei.

[Comment 7] What is the justification for removal of Archaea from the 16S gene dataset. Was the metagenomic data cleaned of archaeal/eukaryote non-fungal/viral sequences to only contain bacterial and fungal reads? Was ORF calling done in such a way as to only include bacterial and fungal sequences?

[Response] In our original MS, we removed archaea from the 16S gene dataset as done by some previous studies using the same primer set (515F and 806R) (DeAngelis *et al.*, 2015; Wagg *et al.*, 2019; Wu *et al.*, 2022). While for the metagenomic data and ORF calling, we didn't remove archaeal/eukaryote non-fungal/viral sequences.

Considering that 515F/806R was universal for both bacterial and archaeal taxa (Barberán *et al.*, 2012; Bates *et al.*, 2011; Yuan *et al.*, 2021), **during our new analyses** (*i.e.*, ASV generation as mentioned in our response to [Comment 13]), we **didn't remove archaea to keep consistent with the metagenomic data**. We have replaced all the results of bacteria with prokaryotes (*i.e.*, bacteria and archaea; Page 21, line 451). Notably, **including archaea did not alter our conclusion based on bacteria** (*i.e.*, warming enhanced prokaryotic network complexity without significant changes in the overall community composition; Fig. 1 and Tables S3-4).

[Comment 8] Ding et al 2017 <https://doi.org/10.1038/ngeo2945>, documents carbon accumulation in the active layers of the QTP, a finding which presumably led to this follow up work to determine processes connected to this accumulation. Please cite this important work and provide more context for the readers and explain why SOC increase was not found in this study.

[Response] Very good comments! We would like to mention that Ding *et al.* (2017) documents carbon accumulation in the 10-30 cm subsurface soil, while **no significant change was detected in the top 10 cm soils**. Our study focused on the 0-10 cm soils without SOC increase, and **thus the result was consistent with that of Ding et al.** (2017).

Nevertheless, we agree with the reviewer that the work of Ding *et al.* (2017) is important in guiding our study on the Tibetan Plateau under climate warming. **We have cited this work in the Introduction section and provided more context for the readers** in the revised MS as follows: “*In such a warming context, an accumulation of SOC was observed in the subsurface soils (10-30 cm) across the plateau, while no significant change was detected for the topsoils (Ding et al., 2017). Yet to date, microbial mechanisms underlying soil C dynamics are largely unexplored on the plateau*” (Page 5, lines 84-87).

[Comment 9] I did not see any section that acknowledges issues or limitations of the methods or design. DNA sequences and their relative abundances do not give absolute values. Absolute values can be obtained by doing quantitative PCR and enzyme assays, or maybe could be linked to the DNA production values in a novel calculation. This project is only one study site, there is therefore no replication other than technical replicas so care needs be taken with extrapolations.

[Response] Very good comments! Following the reviewer’s comments, **we have added a paragraph to discuss the limitations of the method and design in the revised MS** as follows: “*Although this study provided important insights into microbial roles in mediating soil C dynamics in response to experimental warming, some limitations still exist. First, the analyses of microbial community composition and functional capacities were based on DNA sequencing that provided relative abundance data rather than absolute values. Extending studies adopting quantitative polymerase*

chain reaction (PCR), spike-ins of microbial cells (Alteio *et al.*, 2021), and enzyme assays should be conducted to enable the quantification of taxonomic and functional genes. **Second**, the results reported here were derived from one site on the north-eastern Tibetan Plateau. Future research involving more study sites (Fraser *et al.*, 2013; Maes *et al.*, 2024) such as coordinated distributed experiments is required to examine whether the microbial mechanisms underlying soil C processes observed in this grassland ecosystem are applicable to other permafrost regions.” (Pages 14-15, lines 292-302).

Concerns affecting reader interpretation:

[Comment 10] It would be nice to see photos of the field site and the passive warming chambers *in-situ*.

[Response] As suggested by the reviewer, we have added a photo to show the field site and the passive warming chambers *in-situ* (Fig. R3 and Fig. S10).

Fig. R3 Location and design of warming experiment. **a**, Location of the *in situ* warming experiment based on the permafrost distribution map on the Tibetan Plateau (Zou *et al.*, 2017). **b**, Overview of the experimental site. The warming experiment followed a paired design, with ten 4 × 4 m blocks randomly established in a 50 × 50 m fenced area. Photo was taken by Bin Wei. **c**, Design of experimental block. Within each block, a 0.8 × 1.2

m plot located at one corner was arranged as control, and an open-top chamber (OTC) at the diagonally opposite corner was arranged as the warming plot.

[Comment 11] Boxplots are easier for readers to interpret paired comparisons of means rather than bar plots.

[Response] We have replaced the bar plots with boxplots throughout the revised MS.

[Comment 12] Were the same thermal cycle conditions used for both the 16S and ITS reactions?

[Response] We have re-checked the methods of PCR amplification and found that the thermal cycle conditions used for 16S and ITS reactions were not the same (Table R2). We are sorry for the mistake and have provided the right condition in the revised MS (Pages 20-21, lines 431-435).

Table R2 Polymerase chain reaction (PCR) conditions for amplicon sequencing.

Target group	Thermal profile
16S rRNA	94°C for 5 min --× 1 cycle;
	94°C for 30 s, 53°C for 30 s, and 72°C for 30 s --× 30 cycles;
	72°C for 8 min --× 1 cycle
ITS2	98°C for 30 s --× 1 cycle;
	98°C for 10 s, 56°C for 20 s, and 72°C for 30 s --× 32 cycles;
	72°C for 8 min --× 1 cycle

[Comment 13] It is currently more accepted to analyse rRNA gene amplicons using ASVs, and less acceptable to use clustering methods. It would be worthwhile to cite a paper that shows clustering methods, ie OTUs, are as robust a method as ASVs.

[Response] Following the reviewer’s comment and reviewer #2’s [Comment 6] “Consider ASV generation via a denoising technique”, **we have conducted ASV generation using the unoise3 algorithm** (Edgar, 2016), and updated all the results of amplicon sequencing in the revised MS (Page 21, lines 447-448). Notably, **the use of ASVs did not alter our main conclusions based on OTUs**, that is, warming enhanced microbial network complexity without significant changes in the overall microbial community composition for both prokaryotes and fungi (Fig. 1 and Tables S3-4).

[Comment 14] Please consider dividing the number of keystone nodes per network by the number of nodes as the size of the network may bias these results and its more meaningful to see if there are changes in the ratio of keystone to non-keystone OTUs.

[Response] Good suggestion! Following the reviewer's suggestion, we have provided the proportion of the number of keystone nodes to the total nodes in Fig. S3. The results showed that the proportion of keystone nodes was higher for both prokaryotic and fungal networks under warming treatment. Briefly, no keystone node was detected in the two networks under control, while there were 6 (1.5% of total nodes) and 2 (1.7% of total nodes) keystone nodes for prokaryotic and fungal networks under warming treatment. This information has also been added in the revised MS (Page 7, line 138).

[Comment 15] Edaphic and plant variables could be included in the networks to see which modules they were connected to.

[Response] Good comment! The reviewer's comment guided us to explore the connections of edaphic and plant variables with modules in each network based on the eigengene network analysis (Deng *et al.*, 2012; Langfelder & Horvath, 2007; Oldham *et al.*, 2008; Zhou *et al.*, 2011). A module eigengene is a single representative abundance profile that could represent the relative abundance of individual ASVs within the module, and is used to establish relationships with environmental factors by the integrated network analysis pipeline (Feng *et al.*, 2022).

The analyses showed that for prokaryotic network, soil pH, NDVI and root biomass were connected to the major modules under control (Fig. R4a), while soil temperature, pH, NO₃⁻-N content and NDVI were connected to the major modules under the warming treatment (Fig. R4b). For fungal network, environmental factors connected to major modules were soil pH, NO₃⁻-N content, NDVI and root biomass under control (Fig. R4c), and NDVI under warming treatment (Fig. R4d). These results indicated that experimental warming would alter the associations of environmental factors with network modules. We have added these results in the revised MS (Page 7, lines 145-147; Supplementary Note 2, page 6, lines 96-117; Fig. S4).

Fig. R4 Correlations between module eigengene and environmental factors for prokaryotic (a, b) and fungal (c, d) networks under control (a, c) and warming treatment (b, d). Rows correspond to eigengenes of major modules in the network (more than five nodes), whereas columns are the environmental factors. Each plot represents the correlation, of which the color indicating its sign and strength. ST, soil temperature; NDVI, Normalized Difference Vegetation Index. RB, root biomass in 0-10 cm soils. * $P < 0.05$, ** $P < 0.01$.

[Comment 16] No RNA analysis was done so you could not have transcripts per million.

[Response] Following the reviewer's comment, we have removed the use of TPM, and provided the equation for calculating the relative abundance of genes in the revised MS as mentioned in our response to *[Comment 5]* (Pages 22-23, lines 477-481).

[Comment 17] Were the RMT thresholds applied to Pearsons correlation or something else? Please clarify.

[Response] Yes, the RMT thresholds were applied to the Pearson correlation matrix. We have clarified this point in the revised MS (Page 28, lines 591-592).

[Comment 18] Can the networks be plotted with correlation weighted connector lines? Please note that the lines do not represent interactions, they represent correlation which does not prove interaction.

[Response] Following the reviewer’s suggestion, we have plotted the networks with correlation weighted connector lines (Fig. R5). We have also noted that the lines do not prove interaction in the revised MS as follows: “Notably, microbial networks are constructed based on pairwise correlations of ASVs which do not prove direct interactions between microbial taxa, so caution is needed when inferring the associations in these co-occurrence networks (Goberna & Verdú, 2022).” (Page 12, lines 249-252).

Fig. R5 Co-occurrence patterns of topsoil microbial communities as affected by experimental warming. a-d, Visualization of prokaryotic (a, b) and fungal (c, d) networks under control (a, c) and warming (b, d). Nodes in the network denote individual ASVs whose color indicates taxonomic groups. Lines between the nodes represent significant correlations, with yellow and blue indicating positive and negative correlation, respectively. Line width is proportional to the strength of the relationship. n , number of total nodes; L , number of total links; avgK, average degree.

[Comment 19] It is unclear what value plotting networks with module colours adds, is there a more meaningful colour scheme you can use?

[Response] The reviewer's comment guided us to plot the networks with colour scheme of taxonomic groups (*i.e.*, prokaryotic phylum and fungal class as shown in Fig. R5). This colour scheme could be more meaningful because we found alterations in community composition within the networks under experimental warming, and the new colour scheme exactly reflected the taxonomic information of each node.

[Comment 20] Regarding references for node classification: Olesen et al is absolutely a good reference but Guimerà et al is a better second, <https://www.ncbi.nlm.nih.gov/pmc/articles/PMC2175124/> the other references are more about interpretation and so could be removed.

[Response] Following the reviewer's suggestion, we have cited references Olesen *et al.* (2007) and Guimerà *et al.* (2005) for node classification, and removed other references in the revised MS (Page 28, line 599).

[Comment 21] While the node topological classification scheme is respected and well documented, it does look somewhat subjective in Fig 1e,f. Can further justification be given on these choices? Or other choices made?

[Response] Good comments! As mentioned by the reviewer, the use of within-module connectivity (Z_i) and among-module connectivity (P_i) for keystone nodes classification is well documented (Guimerà & Nunes Amaral, 2005; Olesen *et al.*, 2007), and has been widely adopted by previous studies (Shi *et al.*, 2016; Yuan *et al.*, 2021). Of the two parameters, Z_i reflects how well node i is connected to other nodes in the same module, while P_i describes the degree to which node i connects to different modules. Accordingly, nodes in a network could be classified as follows: (i) Connectors ($Z_i \leq 2.5$, $P_i > 0.62$) which “glue” several modules together, (ii) Module hubs ($Z_i > 2.5$, $P_i \leq 0.62$) which are highly connected to many nodes within their own modules, (iii) Network hubs ($Z_i > 2.5$, $P_i > 0.62$) which are important to the coherence of both its own module and the network, and (iv) peripherals ($Z_i \leq 2.5$, $P_i \leq 0.62$) that have a few links within its own module and rarely any to the nodes in other modules (Olesen *et al.*, 2007). **To justify the classification scheme and make it clearer to readers, we have provided more descriptions about the method in the *Supplementary information* (Pages 4-5, lines 73-87).**

The method of Z_i and P_i adopted the same criteria for different networks, based on

which we detected no keystone node for the networks under control. While the results may look subject, **the reviewer’s comment guided us to use other method for keystone classification**. By collecting previous studies, we found that they defined keystone nodes according to node properties such as degree, closeness centrality, and betweenness centrality (Banerjee *et al.*, 2019; Ma *et al.*, 2016; Qiu *et al.*, 2021). Particularly, nodes with high degree are generally considered as keystone nodes in these studies. Thus, we compared node degree between warming and control treatment, and found that **they were significantly different in both prokaryotic and fungal networks** (Fig. R6), confirming the altered node topological roles as revealed by \$Z_i\$ and \$P_i\$ .

Overall, **we considered that the use of Z_i and P_i for keystone node classification was reasonable in our study**. Nevertheless, considering that our conclusion (that is, the microbial networks became more complex under experimental warming) was mainly drawn from the overall network properties such as average degree, average clustering coefficient and density, **we would like to move the results of keystone nodes to the supplementary information to supplement our main findings** (Fig. S3). If the reviewer still considered the results to be subject, we could remove it during the next round of revision. Thanks for your understanding!

Fig. R6 Comparison of node degree between control and warming in prokaryotic (a) and fungal (b) networks. * $P < 0.05$, *** $P < 0.001$ according to Wilcoxon test.

[Comment 22] For all the sequence analyses (whether marker genes or functional genes) all abundances are relative, please fix throughout the manuscript.

[Response] Done as suggested!

[Comment 23] It is unclear what is meant by “functional structure“ in the CAZy section.

[Response] Sorry for the confusion! Functional structure referred to the functional gene composition based on the CAZy families. To avoid the confusion, we have revised the sentence as: “the composition of the CAZy gene families remained unaltered under warming.” (Page 8, lines 159-160).

[Comment 24] The relationship between MBC and CUE was calculated with all data points, it of interest to see if this ratio is different between control and warming (given the caveat of low statistical power with so few data points).

[Response] We calculated the ratio between MBC and CUE, and found that the ratio was not significantly different between control and warming treatment (Fig. R7), suggesting that the relationship between MBC and CUE would not be altered by experimental warming.

Fig. R7 Comparison of MBC/CUE ratio between warming and control conditions according to paired samples t-test (two-sided). Box represents the interquartile range. Horizontal line and circle within the box show the median and mean value, respectively. The whisker denotes SD ($n = 10$). MBC, microbial biomass carbon; CUE, microbial

carbon use efficiency.

[Comment 25] Current best practice with molecular microbiology is to deposit amplicon and metagenome sequences in a public repository/database and supply the accession numbers in the manuscript along with a (private pre-publication) link for the editor/reviewers to verify.

[Response] We have deposited amplicon and metagenome sequences to the NCBI Sequence Read Archive (SRA) with accession number PRJNA1113361 (<https://dataview.ncbi.nlm.nih.gov/object/PRJNA1113361?reviewer=kmqk30qol8u3pc1uvtqag54rp1>). A (private pre-publication) link for the editor/reviewers is provided here, and the data will be released upon publication. We have also provided the accession number in the revised MS (Page 29, lines 619-620).

[Comment 26] It is also current best practice to deposit scripts used for analysis in a public repository with its own citable DOI eg figshare, zenodo, github.

[Response] We have deposited the data and scripts used for analysis in the Figshare data repository (<https://doi.org/10.6084/m9.figshare.25974622.v1>), and provided the information in the revised MS (Page 29, lines 617-618).

Overall, we are very grateful to the reviewer for the insightful comments on our manuscript. These comments guided us to provide more details about the background and methods of the study, and also enabled us to have a deeper thinking on data analyses and representation. By addressing these comments, we feel that the revised MS has been greatly improved. Thank you!

Responses to Reviewer #2

[Comment 1] This manuscript titled “Dual roles of microbes in mediating soil carbon dynamics in response to warming” explores how microbes influence carbon cycle dynamics under experimental warming in permafrost on the Tibetan Plateau using a powerful combination of ecosystem scale and microbial community scale measurements as well as genomic inference. This study helps fill a knowledge gap in permafrost ecology, providing a view into the contrasting ways that microbes impact carbon release and storage. These results are important for future modeling efforts and meta-analyses to understand how climate warming impacts soil microbial carbon cycling.

The authors showed that while microbial community composition was unchanged after 6 years of warming using OTCs, the presence of functional genes indicated increased potential for plant polysaccharide degradation. Microbial community growth and turnover was unchanged, yet heterotrophic respiration, mineral associated carbon, and microbial necromass all increased with warming. These results indicated that microbes are both responsible for the release of carbon through respiration and the stabilization of organic carbon through necromass C production. The suite of tools used in this study and the measurements made were well documented and appropriate for their hypothesis testing.

[Response] Thanks for the reviewer’s positive comments! These comments have guided us to further discuss our findings and update the results of amplicon sequencing. For detailed modifications, please see our responses to the comments listed below.

*[Comment 2] A significant result from this study is that under warming, microbes contributed to soil carbon loss, yet helped stabilize the soil carbon pool by contributing necromass C. However, the authors should clarify how to reconcile unchanged turnover with an increase in necromass C with warming. Increased necromass is controlled by increased microbial growth and turnover (Sokol, N. et al. "Life and death in the soil microbiome: how ecological processes influence biogeochemistry." *Nature Reviews Microbiology* 20.7 (2022): 415-430.) The authors acknowledge this on lines 246-7, that the microbes may be preferentially degrading plant derived compounds, but this fails to explain how necromass C may have increased with unchanged turnover. Please add further interpretation to the discussion. Alternatively, this could be a technical problem since necromass samples were collected in 2019 and growth rate was measured in*

samples from 2020. Can you measure necromass C on the 2020 samples?

[Response] Very good comments! Following the reviewer's suggestion, we firstly measured microbial necromass C for the 2020 samples. The results showed that the total microbial necromass C and its proportion to soil organic carbon (SOC) were both significantly higher under warming treatment (Fig. R8), indicating that our original conclusion (i.e., increased necromass with unchanged turnover) was not a technical problem.

Based on the above-mentioned situation and also following the reviewer's comments, **we have thought more about the reasons for the necromass C accumulation.** Theoretically, the content of microbial necromass in soil is determined by both its input and output, as well as its association with soil minerals (Buckeridge *et al.*, 2020), and thus is a consequence of long-term accumulation. As mentioned by the reviewer and Sokol *et al.* (2022) in their review paper, microbial growth and turnover could affect microbial necromass because they reflect the input of microbial-derived compounds. From this aspect, the unchanged turnover may indicate insignificant difference in necromass formation between warming and control treatment.

Nevertheless, the output and the association of necromass with soil minerals might be altered by experimental warming, which could also lead to changes in necromass over long-term period (that is, six-year warming). **On one hand**, with increased plant C inputs, microorganisms were more likely to utilize plant-derived C as indicated by the elevated relative abundance of related functional genes. Meanwhile, the relative abundance of some genes related to microbial cell wall decomposition was decreased (Fig. 2 and Tables S5). Consequently, microbial residues were relatively less decomposed and thus accumulated over time. **On the other hand**, under the cold environment at our study site, increased soil temperature would favour organo-mineral adsorption reactions like ligand exchange (Conant *et al.*, 2011, Daugherty *et al.*, 2022). As a result, more necromass could be adsorbed to soil minerals and protected from decomposition. Both these two aspects could lead to the accrual of microbial necromass after six-year warming. We have clearly discussed the above-mentioned points in the revised MS (Pages 13-14, lines 268-285). We have also encouraged studies to further explore the possible mechanisms as follows: “*considering that the increase in necromass C is a consequence of long-term accrual, experiments with time-series*

microbial data are encouraged to better understand the alterations in this stable soil C pool.” (Page 14, lines 287-290). Thanks for your understanding!

Fig. R8 Comparison of total microbial necromass C (a) and its proportion to soil organic carbon (SOC; b) between warming and control treatment. Box represents the interquartile range. Horizontal line and circle within the box show the median and mean value, respectively. The whisker denotes SD ($n = 10$). $*P < 0.05$ according to paired samples t-test (two-sided).

Specific points:

[Comment 3] Lines 32-34: add “likely” before “....due to the preferential microbial metabolism....” This study doesn’t provide direct evidence that necromass comes from plant derived compounds.

[Response] Done as suggested!

[Comment 4] Line 156: add something like “potential for” prior to “...enhanced utilization...”

[Response] We have revised the sentence as: “These results collectively indicated enhanced potential for utilization of plant-derived C under warming conditions.” (Page 8, line 169).

[Comment 5] Lines 253-4: Gene presence is only functional potential. Qualify the statement "...indicating that enhanced plant C input by experimental warming would stimulate microbial breakdown of these compounds." With "...experimental warming has the potential to stimulate microbial breakdown..."

[Response] It's true gene presence is only functional potential, and we have reworded the sentence as: "...indicating that with enhanced plant C inputs, experimental warming has the potential to stimulate microbial breakdown of these compounds." (Pages 13-14, lines 279-282).

[Comment 6] Line: OTUs tend to overestimate bacterial enrichment, unless rarefaction and 99% id clustering (Chiarello, M. et al. Ranking the biases: The choice of OTUs vs. ASVs in 16S rRNA amplicon data analysis has stronger effects on diversity measures than rarefaction and OTU identity threshold." PLoS One 17.2 (2022): e0264443.). Consider ASV generation via a denoising technique.

[Response] Following the reviewer's comment, **we have conducted ASV generation using the unoise3 algorithm** (Edgar, 2016), and updated all the results of amplicon sequencing in the revised MS (Page 21, lines 447-448). **The use of ASVs did not alter our main conclusions based on OTUs**, that is, warming enhanced microbial network complexity without significant changes in the overall microbial community composition for both prokaryotes and fungi (Fig. 1 and Tables S3-4).

[Comment 7] Line 404: The Silva138 database was released in 2019 and has important taxonomy updates. This version should be used instead of 132.

[Response] Following the reviewer's comment, we have used Silva138 database to annotate taxonomic information in our new analysis and updated all the results (Page 21, line 449; Fig. S2).

[Comment 8] Line 433: TPM = transcripts per million, but your study didn't analyze gene transcripts.

[Response] We have removed the use of TPM, and provided the equation for calculating the relative abundance of genes in the revised MS (Pages 22-23, lines 477-481).

Overall, we really appreciate for the reviewer's insightful comments, which enabled us

to have a deeper understanding and to be more cautious on the result interpretation. The comments also guided us to analyze our data with more proper methods. By addressing these comments, we feel that the revised MS has been greatly improved. Thank you!

References

- Alteio LV, S neca J, Canarini A *et al.*, 2021. A critical perspective on interpreting amplicon sequencing data in soil ecological research. *Soil Biology and Biochemistry*, **160**, 108357.
- Banerjee S, Walder F, B chi L *et al.*, 2019. Agricultural intensification reduces microbial network complexity and the abundance of keystone taxa in roots. *The ISME Journal*, **13**, 1722-1736.
- Barber n A, Bates ST, Casamayor EO, Fierer N, 2012. Using network analysis to explore co-occurrence patterns in soil microbial communities. *The ISME Journal*, **6**, 343-351.
- Bates ST, Berg-Lyons D, Caporaso JG *et al.*, 2011. Examining the global distribution of dominant archaeal populations in soil. *The ISME Journal*, **5**, 908-917.
- Bokhorst S, Huiskes A, Aerts R *et al.*, 2013. Variable temperature effects of Open Top Chambers at polar and alpine sites explained by irradiance and snow depth. *Global Change Biology*, **19**, 64-74.
- Buckeridge KM, Mason KE, McNamara NP *et al.*, 2020. Environmental and microbial controls on microbial necromass recycling, an important precursor for soil carbon stabilization. *Communications Earth & Environment*, **1**, 36.
- Cheng J, Yang Y, Yuan MM *et al.*, 2021. Winter warming rapidly increases carbon degradation capacities of fungal communities in tundra soil: potential consequences on carbon stability. *Molecular ecology*, **30**, 926-937.
- Conant RT, Ryan MG,  gren GI *et al.*, 2011. Temperature and soil organic matter decomposition rates - synthesis of current knowledge and a way forward. *Global Change Biology*, **17**, 3392-3404.
- Daugherty EE, Lobo GP, Young RB *et al.*, 2022. Temperature effects on sorption of dissolved organic matter on ferrihydrite under dynamic flow and batch conditions. *Soil Science Society of America Journal*, **86**, 224-237.
- DeAngelis KM, Pold G, Top uođlu BD *et al.*, 2015. Long-term forest soil warming alters microbial communities in temperate forest soils. *Frontiers in Microbiology*, **6**, 104.
- Deng Y, Jiang Y-H, Yang Y *et al.*, 2012. Molecular ecological network analyses. *BMC*

- Bioinformatics*, **13**, 113.
- Ding J, Chen L, Ji C *et al.*, 2017. Decadal soil carbon accumulation across Tibetan permafrost regions. *Nature Geoscience*, **10**, 420-424.
- Dorrepaal E, Toet S, van Logtestijn RSP *et al.*, 2009. Carbon respiration from subsurface peat accelerated by climate warming in the subarctic. *Nature*, **460**, 616-619.
- Edgar RC, 2016. UNOISE2: improved error-correction for Illumina 16S and ITS amplicon sequencing. *bioRxiv*, 081257.
- Facelli JM, Pickett STA, 1991. Plant litter: light interception and effects on an old-field plant community. *Ecology*, **72**, 1024-1031.
- Feng K, Peng X, Zhang Z *et al.*, 2022. iNAP: an integrated network analysis pipeline for microbiome studies. *iMeta*, **1**, e13.
- Fraser LH, Henry HA, Carlyle CN *et al.*, 2013. Coordinated distributed experiments: an emerging tool for testing global hypotheses in ecology and environmental science. *Frontiers in Ecology and the Environment*, **11**, 147-155.
- Goberna M, Verdú M, 2022. Cautionary notes on the use of co-occurrence networks in soil ecology. *Soil Biology and Biochemistry*, **166**, 108534.
- Guimerà R, Nunes Amaral LA, 2005. Functional cartography of complex metabolic networks. *Nature*, **433**, 895-900.
- Hasselquist NJ, Metcalfe DB, Högberg P, 2012. Contrasting effects of low and high nitrogen additions on soil CO₂ flux components and ectomycorrhizal fungal sporocarp production in a boreal forest. *Global Change Biology*, **18**, 3596-3605.
- Khan MAW, Bohannan BJM, Nüsslein K *et al.*, 2019. Deforestation impacts network co-occurrence patterns of microbial communities in Amazon soils. *Fems Microbiology Ecology*, **95**, fiy230.
- Kuzyakov Y, 2006. Sources of CO₂ efflux from soil and review of partitioning methods. *Soil Biology and Biochemistry*, **38**, 425-448.
- Langfelder P, Horvath S, 2007. Eigengene networks for studying the relationships between co-expression modules. *BMC Systems Biology*, **1**, 54.
- Ma B, Wang H, Dsouza M *et al.*, 2016. Geographic patterns of co-occurrence network topological features for soil microbiota at continental scale in eastern China. *The ISME Journal*, **10**, 1891-1901.
- Maes SL, Dietrich J, Midolo G *et al.*, 2024. Environmental drivers of increased ecosystem respiration in a warming tundra. *Nature*, **629**, 105-113.

- Nottingham AT, Meir P, Velasquez E, Turner BL, 2020. Soil carbon loss by experimental warming in a tropical forest. *Nature*, **584**, 234-237.
- Oldham MC, Konopka G, Iwamoto K *et al.*, 2008. Functional organization of the transcriptome in human brain. *Nature Neuroscience*, **11**, 1271-1282.
- Olesen JM, Bascompte J, Dupont YL, Jordano P, 2007. The modularity of pollination networks. *Proceedings of the National Academy of Sciences of the United States of America*, **104**, 19891-19896.
- Qin J, Li Y, Cai Z *et al.*, 2012. A metagenome-wide association study of gut microbiota in type 2 diabetes. *Nature*, **490**, 55-60.
- Qiu L, Zhang Q, Zhu H *et al.*, 2021. Erosion reduces soil microbial diversity, network complexity and multifunctionality. *The ISME Journal*, **15**, 2474-2489.
- Shi S, Nuccio EE, Shi ZJ *et al.*, 2016. The interconnected rhizosphere: high network complexity dominates rhizosphere assemblages. *Ecology Letters*, **19**, 926-936.
- Wagg C, Schlaeppi K, Banerjee S *et al.*, 2019. Fungal-bacterial diversity and microbiome complexity predict ecosystem functioning. *Nature Communications*, **10**, 4841.
- Wu L, Yang F, Feng J *et al.*, 2022. Permafrost thaw with warming reduces microbial metabolic capacities in subsurface soils. *Molecular ecology*, **31**, 1403-1415.
- Yuan MM, Guo X, Wu L *et al.*, 2021. Climate warming enhances microbial network complexity and stability. *Nature Climate Change*, **11**, 343-348.
- Zhou J, Deng Y, Luo F *et al.*, 2011. Phylogenetic molecular ecological network of soil microbial communities in response to elevated CO₂. *mBio*, **2**, e00122-11.
- Zhou Y, Sun B, Xie B *et al.*, 2021. Warming reshaped the microbial hierarchical interactions. *Global Change Biology*, **27**, 6331-6347.
- Zou D, Zhao L, Sheng Y *et al.*, 2017. A new map of permafrost distribution on the Tibetan Plateau. *The Cryosphere*, **11**, 2527-2542.

REVIEWERS' COMMENTS

Reviewer #2 (Remarks to the Author):

The authors did a thorough job addressing reviewer concerns. They should be proud of their hard work and contribution to the field of microbial soil C cycling under environmental change. Some minor comments below.

Line 130: “between warming and control treatment” change to “between the warming and control treatments”. And “Compared with control, warming treatment”, change to “Compared to the control, the warming treatment...” When referring to “x treatment”, you need an article (typically “the”). This occurs in multiple lines in the manuscript, please correct. Sometimes you can simply remove the word “treatment”, for instance on line 163 and 196 “...under warming treatment” can be “...under warming...”

Line 233: change “after more than half-decade warming treatment” to “after more than a half-decade of warming...”

Line 172: ITS isn't a gene, please change to “16S rRNA gene and ITS2 region”

Line 333 and 346 and in multiple methods sections: put a space before the degrees C symbol °C

Line 457: change “We obtained DNAs for the 20 soil samples...” to “We performed metagenomic sequencing on 20 soil samples to explore...”

Line 477: by “gene catalogue” do you mean your metagenome coassembly? Please clarify

Fig 2 panel a - the y axis is relative abundance, but the scale is 10⁴ - Usually a relative abundance sums to 1. Please adjust.

Responses to Reviewer #2

[Comment 1] *The authors did a thorough job addressing reviewer concerns. They should be proud of their hard work and contribution to the field of microbial soil C cycling under environmental change. Some minor comments below.*

[Response] Thanks for the reviewer's insightful comments! We have addressed the reviewer's comments and made revisions in the manuscript. For detailed modifications, please see our responses to the comments listed below.

[Comment 2] *Line 130: “between warming and control treatment” change to “between the warming and control treatments”. And “Compared with control, warming treatment”, change to “Compared to the control, the warming treatment...” When referring to “x treatment”, you need an article (typically “the”). This occurs in multiple lines in the manuscript, please correct. Sometimes you can simply remove the word “treatment”, for instance on line 163 and 196 “...under warming treatment” can be “...under warming...”*

[Response] Following the reviewer's comments, we have checked throughout the manuscript, and added “the” before “x treatment” (Page 7, lines 126 and 138; Page 9, line 174). We have also removed the word “treatment” in some sentences as suggested by the reviewer.

[Comment 3] *Line 233: change “after more than half-decade warming treatment” to “after more than a half-decade of warming...”*

Line 172: ITS isn't a gene, please change to “16S rRNA gene and ITS2 region”

Line 333 and 346 and in multiple methods sections: put a space before the degrees C symbol °C

Line 457: change “We obtained DNAs for the 20 soil samples...” to “We performed metagenomic sequencing on 20 soil samples to explore....”

[Response] Done as suggested.

[Comment 4] *Line 477: by “gene catalogue” do you mean your metagenome coassembly? Please clarify*

[Response] Following the reviewer's suggestion, we have provided more details about the processes of assembly, and clearly defined gene catalogue in the revised MS (Page 22, lines 458-460 and 463).

[Comment 5] Fig 2 panel a - the y axis is relative abundance, but the scale is 104 - Usually a relative abundance sums to 1. Please adjust.

[Response] We have adjusted the y axis to the values calculated by the equation 1 as provided in the manuscript to make the information more clear.